# Charting and probing the activity of ADARs in human development and cell-fate specification

Amir Dailamy[1,6], Weiqi Lyu[1,2,6], Sami Nourreddine[1], Michael Tong[1], Joseph Rainaldi[1,3], Daniella McDonald[1,3], Rebecca Panwala[1], Alysson Muotri[4], Michael S. Breen[5], Kun Zhang[1,2] & Prashant Mali [1] ✉

Adenosine deaminases acting on RNA (ADARs) impact diverse cellular processes and pathological conditions, but their functions in early cell-fate specification remain less understood. To gain insights here, we began by charting time-course RNA editing profiles in human organs from fetal to adult stages. Next, we utilized hPSC differentiation to experimentally probe ADARs, harnessing brain organoids as neural specific, and teratomas as pan-tissue developmental models. We show that time-series teratomas faithfully recapitulate fetal developmental trends, and motivated by this, conducted pan-tissue, single-cell CRISPR-KO screens of ADARs in teratomas. Knocking out ADAR leads to a global decrease in RNA editing across all germ-layers. Intriguingly, knocking out ADAR leads to an enrichment of adipogenic cells, revealing a role for ADAR in human adipogenesis. Collectively, we present a multi-pronged framework charting time-resolved RNA editing profiles and coupled ADAR perturbations in developmental models, thereby shedding light on the role of ADARs in cell-fate specification.

Adenosine-to-inosine (A-to-I) editing is a critical post-transcriptional modification that alters RNA nucleotides, is catalyzed by adenosine deaminases acting on RNA (ADAR) enzymes, and is one of the most prominent epitranscriptomics modification in metazoans[1–8]. There are 3 human ADAR enzymes: *ADAR*, *ADARB1*, and *ADARB2*. *ADAR* (also known as *ADAR1*) is constitutively expressed across most tissues, and is mostly known to edit repetitive RNA[9]. *ADARB1* (also known as *ADAR2*) is also expressed in most tissues, with the highest expression in the CNS, and known to mostly edit coding regions[9,10]. *ADARB2* (also known as *ADAR3*) is exclusively expressed in the CNS and contains a catalytically inactive deaminase domain; thus, it does not edit RNA and is mostly known to play a regulatory role in RNA editing[9,11]. Adenosines are deaminated and converted to inosines in A-to-I RNA editing, and the inosines are interpreted as guanosines by cellular chemistry.

Consequently, A-to-I editing can yield unique protein isoforms[10,12,13] and also influence miRNA[14] and mRNA stability[15], splicing[16], and localization[17]. Furthermore, the dysregulation of ADAR proteins, both via their RNA editing-dependent and -independent[18,19] functions, have been linked to various diseases, such as cancer[20], neurological disorders[21,22], viral infection[23], metabolic disorders[24], and auto-immune malignancies[25].

Systematic investigations into RNA editing profiles in adult human tissue have documented editing sites across tissues and uncovered key regulators in RNA editing[13,26,27]. In addition to their roles in adult tissues, ADAR enzymes are pivotal regulators of post-transcriptional RNA processing during various developmental stages[27–31]. In mice, *Adar*-KO[18,32,33] and *Adarb1*-KO[34,35] result in embryonic and perinatal lethality, respectively, highlighting their putative role in developmental

[1]Department of Bioengineering, University of California San Diego, San Diego, CA, USA. [2]Altos Labs, San Diego, CA, USA. [3]Biomedical Sciences Graduate Program, University of California San Diego, San Diego, CA, USA. [4]Department of Pediatrics and Cellular & Molecular Medicine, University of California San Diego, San Diego, CA, USA. [5]Icahn School of Medicine at Mount Sinai, New York, NY, USA. [6]These authors contributed equally: Amir Dailamy, Weiqi Lyu. ✉e-mail: pmali@ucsd.edu

processes. However, due to these lethality phenotypes, mouse models are limited in their capacity to systematically study ADAR's role in cell fate specification. Lastly, given the prevalence of A-to-I editing in primate-specific *Alu* elements[36], mouse models may not capture the human-specific roles that ADARs have in human development.

Taken together, while significant insights exist regarding ADARs' roles in mouse models and adult human tissues, a knowledge gap remains concerning their impact on early human organogenesis. Towards this, we established here a multi-pronged framework (Fig. 1) to systematically map and functionally interrogate their role in human development. We first charted time-course RNA editing profiles in five different human organ tissues, from fetal to adult stages. Next, to experimentally probe ADARs, we utilized two human pluripotent stem cell (hPSC) derived models of development: cerebral organoids, in vitro derived brain tissue-specific constructs; and teratoma tissues, in vivo derived pan-tissue constructs[37]. The multilineage nature of the teratoma allowed the analysis of RNA editing across all germ-layers in a single experiment, motivating a deeper investigation into teratoma tissue as a developmental model for RNA editing. By generating time-series teratomas, we observed that neural tissue in teratomas replicates transcriptomic and epitranscriptomic developmental patterns evident in fetal cerebral tissue. Moreover, a pan-tissue, single-cell CRISPR-KO screen in teratomas unveiled *ADAR*'s integral role across all germ layers, with a distinct emphasis on adipogenic cell-fate determination, suggesting *ADAR*'s potential implication in obesity-related phenotypes.

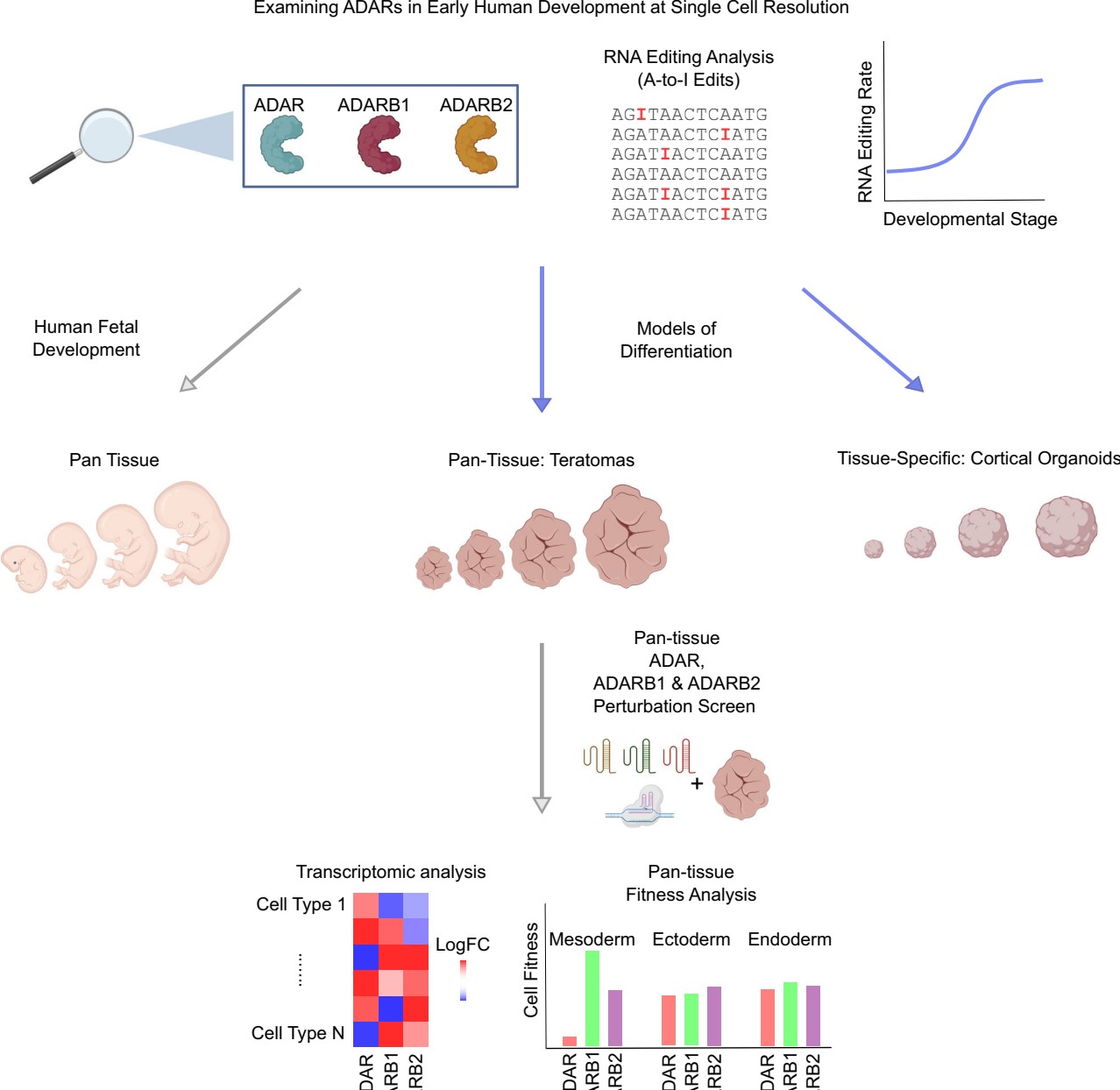

**Fig. 1 | This graphical abstract illustrates our study's workflow.** We begin by analyzing RNA editing and ADAR expression dynamics across diverse tissue types, utilizing both human organ data and hPSC-derived developmental models. Subsequently, we employed the teratoma for a pan-tissue ADAR perturbation screen, probing the functional roles of ADAR proteins across all three germ layers.

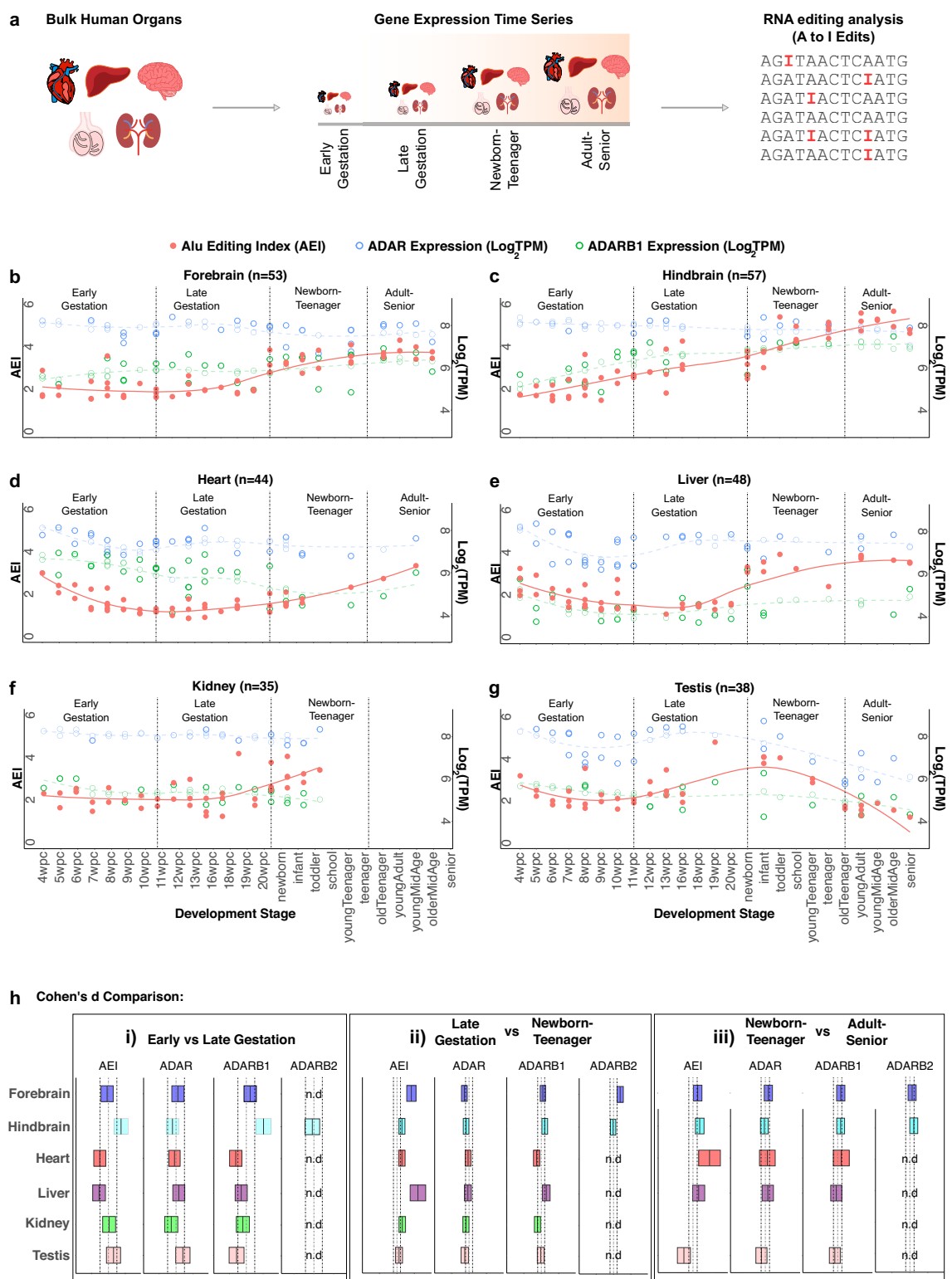

**Fig. 2 | Time series bulk RNA editing analysis in human fetal tissues. a** Schematic depicting the workflow for this investigation into RNA editing and ADAR expression dynamics in bulk organ tissues across lifespan. Organ time-series data is binned into 4 developmental groups—early gestation (4–10 wpc), late gestation (11–20 wpc), newborn-to-teenager (0–19 years of age), and adult-to-senior (25–63 years of age)—for conducting comparisons across sequential developmental periods. **b**–**g** AEI levels (solid red circles), as well as *ADAR* (open blue circles) and *ADARB1* (open green circles) expression values, are charted throughout lifespan for the forebrain (**b**), hindbrain (**c**), heart (**d**), liver (**e**), kidney (**f**), and testis (**g**). **h** Cohen's *d* comparison between sequential time groups for AEI, *ADAR*, *ADARB1*, and *ADARB2*, across all organs (not detected = n.d, box represents 95% confidence interval).

This study provides a pan-tissue, single-cell-level investigation into the spatiotemporal profiles and functional roles of ADARs during human fetal development. By investigating a range of fetal and fetal-like tissues, we elucidate the dynamic RNA editing landscape, casting light on its implications in human development and disease.

## Results

### RNA editing analysis in bulk human organs

We first employed a time-resolved mammalian gene expression database[38] to systematically investigate temporal RNA editing levels in multiple human organs, including the forebrain, hindbrain, heart, liver, kidney and testis. We categorized the data into four developmental stages: early gestation (4–10 weeks post conception, wpc), late gestation (11–20 wpc), newborn-teenager (0–19 years old), and adult-senior (25–63 years old) (Fig. 2a).

A-to-I editing events are predominantly observed in *Alu* elements—repetitive DNA sequences exclusive to the primate genome[36]. Given its robustness, and especially as it accounts for the editing activity in low covered regions, while avoiding the need to quantify the editing level per-site, we assessed the global editing levels in bulk human organs via the Alu editing index (AEI)[39]. This is obtained by calculating the ratio of A-to-G mismatches to the total adenosine coverage within Alu regions. We then measured the AEI shifts and ADAR enzyme expression dynamics across successive developmental stages.

In the forebrain, we observed a significant rise in AEI (p-value = 1.24E-10, Cohen's D = 4.12) during the late gestation to the newborn-teenager transition, accompanied by a decrease in *ADAR* (p-value = 2.54E-3, Cohen's D = −1.35) and an increase in *ADARB2* (*p*-value = 1.86E-05, Cohen's D = 1.91) (Fig. 2b). Furthermore, in the hindbrain we observe an upward AEI trend throughout all of lifespan (Fig. 2c). We observe concordant significant increases between both the AEI and *ADARB1* expression in the early-to-late gestation transition – AEI (p-value = 7.72E-04, Cohen's D = 1.50) and *ADARB1* (p-value = 1.17E-06, Cohen's D = 2.04)—and from late gestation to the newborn-teenager transition—AEI (p-value = 4.57E-04, Cohen's D = 1.42) and *ADARB1* (p-value = 1.50E-02, Cohen's D = 0.99)—which suggests *ADARB1* is driving the increase in AEI during hindbrain development (Fig. 2c). These findings are consistent with a recent report tracking RNA editing in developing human brain tissue[27].

In the heart, we observe pronounced AEI fluctuations throughout development (Fig. 2d). Initially, in the early-to-late gestation transition, there is a notable drop in AEI (p-value = 2.73E-03, Cohen's D = −1.03), followed by an increase in AEI during the late gestation to newborn-teenager transition (p-value = 4.56E-04, Cohen's D = 1.43) (Fig. 2d). This initial decrease and subsequent increase in AEI are concordant with drops in *ADARB1* expression during both the early-to-late gestation transition (p-value = 2.15E-03, Cohen's D = −1.35) and the late gestation to newborn-teenager transition (p-value = 4.56E-04, Cohen's D = 1.43) (Fig. 2d).

The liver exhibited very dynamic shifts in AEI across lifespan (Fig. 2e), with an initial decrease in AEI during the early-to-late gestation transition (p-value = 3.21E-04, Cohen's D = −1.11), a robust increase in AEI during the late gestation to the newborn-teenager transition (p-value = 3.28E-05, Cohen's D = 6.00), and finally another significant increase in AEI during the newborn-teenager to adult-senior transition (p-value = 4.40E-02, Cohen's D = 1.26) (Fig. 2e). Again, the initial drop and subsequent rise in AEI were concordant with decreases (p-value = 4.83E-02, Cohen's D = −0.60) and subsequent increases (p-value = 4.36E-02, Cohen's D = 1.43) in *ADARB1* expression, respectively (Fig. 2e).

In the kidney, we observed a significant increase in AEI during the late gestation to newborn-teenager transition (p-value = 4.77E-04, Cohen's D = 1.51), coinciding with a significant drop in *ADARB1* (p-value = 1.25E-02, Cohen's D = −1.10) (Fig. 2f).

In the testis we observed a robust AEI reduction (p-value = 9.94E-05, Cohen's D = −5.21) during the newborn-teenager to adult transition, which is concordant with a robust and significant decrease in *ADAR* expression (p-value = 3.33E-03, Cohen's D = -2.19) (Fig. 2g, h).

Intrigued by a notable drop in heart and liver AEI during early gestation (Fig. 2d, e), we further investigated this trend by dividing the early gestation period into two phases: 4–7 wpc and 8–11 wpc. During this developmental shift, we observed a significant reduction in AEI in heart tissue (p-value = 4.35E-03, Cohen's D = −1.80), which is concordant with a significant decrease in *ADAR* (p-value = 1.81E-05, Cohen's D = −2.64). In the liver, we also noted a substantial decrease in AEI (p-value = 2.13E-02, Cohen's D = −1.00), accompanied by significant reductions in both *ADAR* (p-value = 1.79E-04, Cohen's D = −1.82) and *ADARB1* (p-value = 9.75E-3, Cohen's D = −1.24) expression.

In summary, there are few convergent trends with respect to AEI and ADAR expression dynamics across tissues. *ADARB1* expression dynamics closely follows the progressive increase in AEI throughout hindbrain development, and a drop in AEI during very early gestation is accompanied by a drop in *ADAR* expression in both heart and liver tissue. After examining RNA editing trends throughout human development in bulk tissues, we next shifted our attention to a single-cell investigation of RNA editing at prenatal stages to uncover cell-type specific trends that may govern early organogenesis. To explore this, we employed lab-grown, hPSC-derived model systems, complemented with human fetal tissue data[40].

### Site-specific editing analysis in bulk human organs

To gain a deeper insight into A-to-I editing dynamics beyond just global editing metrics, we employed a site-specific RNA editing pipeline (Supplementary Fig. 1A)[27]. This pipeline identified edited mRNA sites within the REDIportal table across all bulk organ time-series data (forebrain, hindbrain, liver, heart, kidney, and testis). Subsequently, we examined differential editing levels in these sites between postnatal and prenatal samples using customized R scripts, akin to established computational methodologies[27]. The resulting delta editing values (differential editing rate between postnatal and prenatal samples, as defined in previous applications[27]) were then correlated with the log-fold change in expression levels of corresponding genes between postnatal and prenatal samples. After identifying RNA editing sites at a site-specific level, we next classified editing sites into their corresponding genetic regions, and found that the majority of identified sites are located in 3'UTRs (Supplementary Fig. 1B). Furthermore, we found that the largest number of editing sites were identified in the hindbrain (Supplementary Fig. 1B).

Next, we identified sites that are differentially edited between prenatal and postnatal samples across all tissues (Supplementary Fig. 1C). The delta editing rate of these differentially edited sites was then correlated with the log fold change of the corresponding gene expression level changes. Across all organs, we observed a slightly negative correlation (R = −0.133) between the change in editing levels and gene expression across all identified differentially edited genes (Supplementary Fig. 1D).

In order to shed light on potential developmentally conserved RNA editing mechanisms, we next aimed to uncover genes with differentially edited sites common to multiple tissue types (Fig. 3a). We observed 58 prenatal versus postnatal differentially edited sites that were shared across all organ datasets (Fig. 3b). Intrigued by this convergence, we conducted functional annotation on these genes using the ToppGene Suite and identified GO terms associated with innate immunity and DNA replication (Fig. 3c). Notably, *EIF2AK2* (also known as *PKR*) and *MAVS*, involved in viral response and innate immunity pathways, were found to have differentially edited sites in all analyzed tissues (Fig. 3d). Thus, through an unbiased look at the convergence of prenatal versus postnatal differentially edited RNA editing sites across organs from all three germ-layers, we identified that there is a

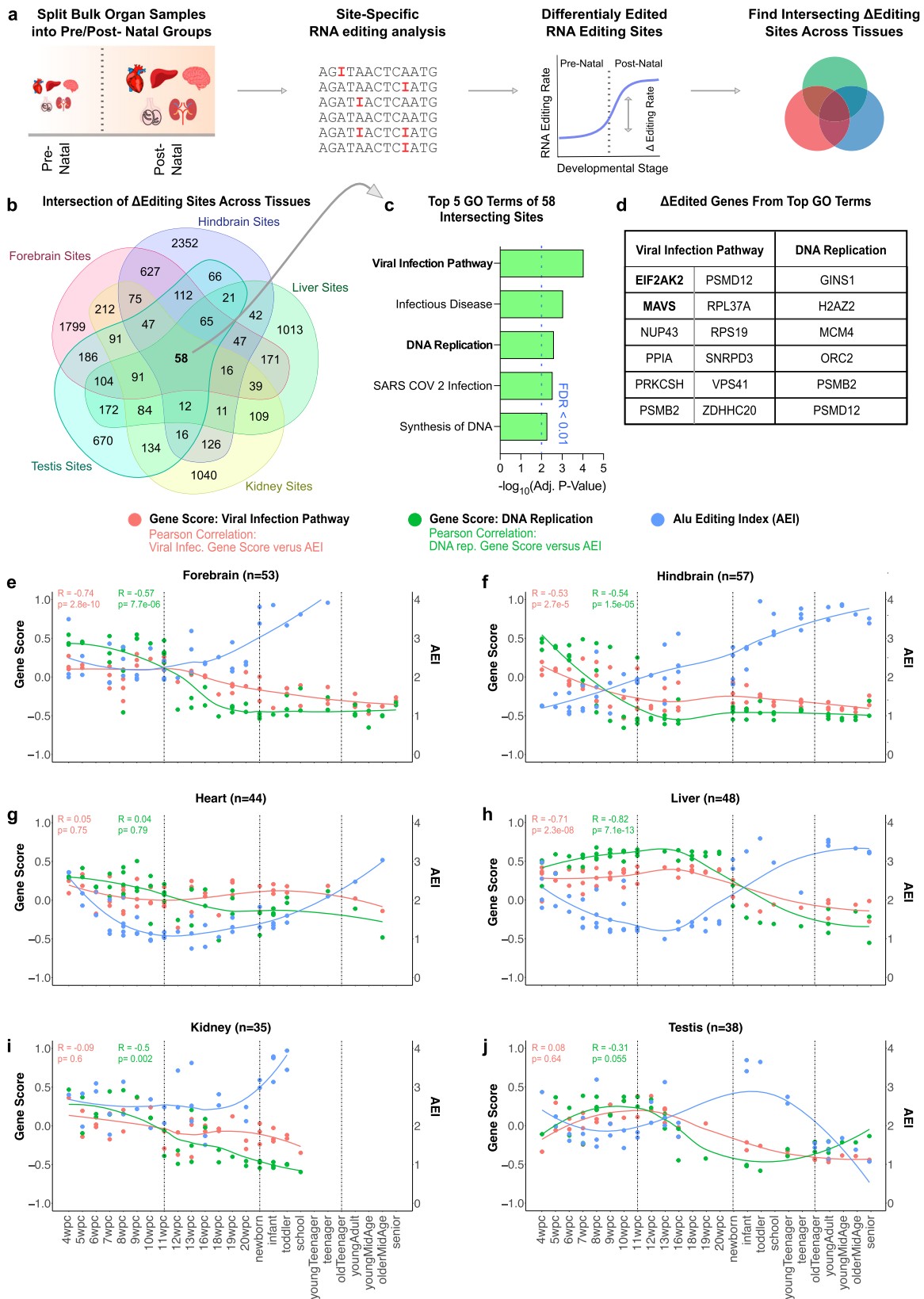

significant shift in RNA editing of innate immunity-related genes during the fetal-to-adult transition.

To assess how shifts in RNA editing correlate with gene expression, we utilized Gene Set Variation Analysis (GSVA)[41], which condenses information from gene expression profiles to calculate gene scores for specific biological pathways. Specifically, we computed gene scores for C2 gene sets from the Human MSigDB Collections[42,43], tracking them over time for the Viral Infection Gene Set (red line) and DNA Replication Gene Set (green line) (Fig. 3e–j). The global editing level of these samples, measured by the Alu Editing Index (AEI, blue line), was overlaid onto the graphs to visualize changes in global editing levels across development for each organ (Fig. 3e–j). We

**Fig. 3 | Examining the convergence of differentially edited sites across organ systems. a** Schematic depicting the site-specific RNA editing analysis pipeline for developmental bulk organ samples. After bulk organ samples were grouped into prenatal and postnatal groups, differential editing analysis was conducted between these two developmental groups. Finally, an analysis was conducted to find shared and unique differentially edited sites between organs. **b** Venn diagram illustrating the number of shared and unique prenatal versus postnatal differentially edited

sites across all examined organ datasets. **c** GO term functional annotation of the pan-organ shared differentially edited sites. **D** List of the input genes that are associated with these corresponding convergent GO terms. **e–j** Charting global A-to-I editing dynamics (AEI), along with the Viral Infection and DNA replication pathway gene scores, across developmental time, for the forebrain (**e**), hindbrain (**f**), heart (**g**), liver (**h**), kidney (**i**), and testis (**j**). Correlation between the gene score sets and the AEI was measured using the Pearson correlation coefficient.

observed significant inverse correlations between the AEI and gene scores from these two pathways in the forebrain (Fig. 3e), hindbrain (Fig. 3f), and liver (Fig. 3h). This analysis underscores the crucial role of A-to-I editing in regulating gene expression in innate immunity and DNA replication pathways during the fetal-to-adult transition.

### Single-cell RNA editing analysis in hPSC-derived model systems

After exploring A-to-I editing dynamics in adult bulk tissue, we sought to explore A-to-I editing trends in early human tissue development at a single-cell level, starting with 8-week-old cerebral organoids and 8–10-week-old teratomas, both derived from human pluripotent stem cells (Fig. 4a). As compared to bulk RNA sequencing, single-cell RNA sequencing data is characterized by more limited coverage and a 3' bias due to the poly-A based capture probes utilized via the 10X Chromium platform[44]. Thus, we decided to utilize the robust AEI index to more accurately estimate global editing levels.

We started our investigation with cerebral organoids, and observed that 8-week-old cerebral organoids were composed of 3 main cell types: Radial Glia, Intermediate Progenitors, and Glutamatergic Neurons (Fig. 4b). We measured AEI trends across these cell types (Fig. 4c) as well as the correlation between AEI and expression of *ADAR* and *ADARB1* for each cell type (Fig. 4d). To gain insight into the pan-tissue influence of ADARs, we next considered the teratoma as an additional model system. Among neural lineages, we observed 8–10 week-old teratomas included Radial Glia, Intermediate Progenitors, Glutamatergic Neurons and also GABAergic neurons, and across all tissues composed of over 20 distinct cell-types representative of all three germ layers (Fig. 4e)[37]. We confirmed these cell-types exhibit a strong correlation with their corresponding fetal counterparts (Supplementary Fig. 2A). Moreover, we assessed the reliability of teratoma single-cell RNA sequencing libraries for RNA editing analysis by comparing single-cell editing rates with those from bulk teratoma RNA sequencing libraries, revealing a high correlation between these two library generation and sequencing modalities (Supplementary Fig. 2B). Finally, we confirmed expression of key marker genes (Supplementary Fig. 2C). As expected, *ADAR* exhibited ubiquitous expression across all teratoma cell-types, while *ADARB2* was neural-specific and *ADARB1* was most highly expressed in neural tissues, and at lower levels elsewhere (Supplementary Fig. 2D). Lastly, we conducted a similar teratoma generation pipeline on 3 other commonly used PSC cell lines (H9s, HUES62s, and PGP1s, Supplementary Fig. 3A). We were able to call all major cell-types in H9 teratomas (Supplementary Fig. 3B) and saw similar AEI levels (Supplementary Fig. 3C) and AEI-to-ADAR correlation levels (Supplementary Fig. 3D) as well. These observations and trends were also seen in the HUES62 (Supplementary Fig. 3E-G) and PGP1 (Supplementary Fig. 3H–J) cell lines.

To quantify RNA editing levels in unique teratoma cell-types, we created pseudo-bulk samples by pooling single cell RNA count matrices from matching cell-types together (refer "Methods"). Our observations revealed varying levels of AEI across teratoma cell types (Fig. 4f). Notably, we observe a significantly lower AEI in muscle cells compared to the average teratoma AEI (Fig. 4f), which is consistent with reports examining RNA editing across adult human bulk tissue samples[9,39]. Furthermore, *ADAR* and *ADARB1* expression explained 31% and 10% of the variance in teratoma cell-type AEI, respectively (Fig. 4g), which is also consistent with adult human bulk tissue sample reports[9].

In summary, the teratoma presents a promising model for RNA editing analysis due to its rich panoply of cell-types spanning all three germ-layers and its notable AEI-to-ADAR correlation trends among these cell-types. Importantly, generating the teratoma only required 8–10 weeks compared to the 5 months needed to generate cerebral organoids with comparable neural cell-type diversity[45]. Given these favorable characteristics, we decided to proceed with the teratoma for our subsequent investigations.

### RNA editing analysis in time series teratomas

To investigate temporal RNA editing trends in teratoma cell types and determine their correlations with human fetal tissue, we produced teratomas at four distinct developmental stages: 4, 6, 8, and 10 weeks. Given that we would be harvesting and storing our samples in OCT blocks as they reach the desired developmental states (4, 6, 8, and 10 weeks), and subsequently processing RNAseq libraries all at once, we sought a library preparation method that would be broadly applicable and compatible with this storage modality. Towards this, we explored single-nucleus RNA sequencing (snRNAseq) as it is compatible with many methods of sample preservation, including frozen OCT blocks. To assess if both library processing modalities were comparable in the context of RNA editing, we first compared RNA editing metrics and cell recovery rates across platforms. Although there are differences in total number of captured RNA editing sites (Supplementary Fig. 4A, B) and cell recovery rates (Supplementary Fig. 4C), RNA editing rates across single cell and single nucleus libraries correlate strongly (R = 0.811) with one another (Supplementary Fig. 4D). Motivated by the concordance across modalities, we then constructed single nucleus RNA sequencing (snRNA-seq) libraries, and conducted transcriptomic and RNA editing analysis across these samples (Fig. 5a).

First, we confirmed that teratomas from the same developmental stage strongly correlated with each other (Supplementary Fig. 5A). We then examined how cell-type proportions change during teratoma development. Notably, early-stage teratomas exhibited a higher proportion of germ-layer progenitors, while later-stage teratomas were predominantly composed of their differentiated progeny (Supplementary Fig. 5B-F). For instance, in the ectoderm, week 4 teratomas were primarily composed of radial glia, constituting about 65% of the ectodermal cell types, while this proportion gradually decreased to ~35% in week 10 teratomas (Supplementary Fig. 5C). Conversely, the relative proportion of early neurons in the ectoderm was less than 10% in week 4 teratomas, gradually increasing to around 40% in week 10 teratomas (Supplementary Fig. 5C). We observed similar trends in the endoderm, where precursor definitive endoderm cells in early-stage teratomas gave rise to mid/hindgut epithelium in the later stage teratomas (Supplementary Fig. 5D); and also in the mesoderm, where COL15A positive MSC/Fibroblast progenitors[46] (termed here as MyoFib cells) exhibit a high initial relative mesodermal percentage that progressively decreases during teratoma development (Supplementary Fig. 5F). These results demonstrate that teratoma cell types reproducibly mature into their differentiated progeny in a progressive manner.

Similar to week 10 teratomas (Supplementary Fig. 2D), we observed ubiquitous expression of *ADAR* across all stages of teratoma development, while *ADARB2* exhibited neural-specific expression, and *ADARB1* showed strongest expression in neural tissues but also was expressed at low levels elsewhere (Supplementary Fig. 6A-C).

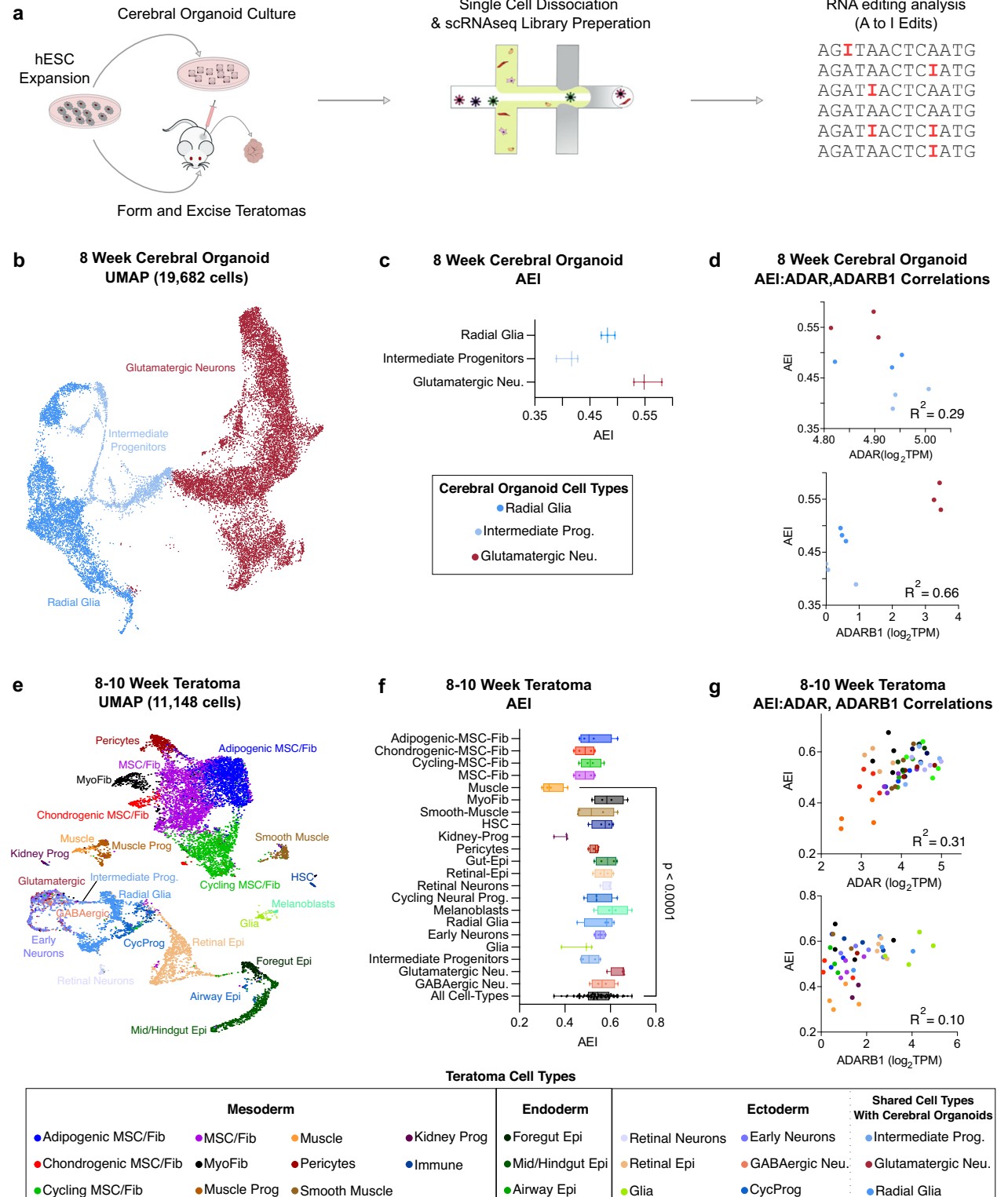

**Fig. 4 | Single-cell RNA editing analysis in hESC-derived model systems.**
**a** Workflow schematic for conducting single-cell RNA editing analysis on hESC-derived cerebral organoids and teratomas. **b** UMAP plot from 8-week cerebral organoids processed through the single-cell RNA sequencing pipeline. **c** AEI values for all major 8-week cerebral organoid cell-types, with each data point calculated as a pseudo-bulk value from each cerebral organoid cell-type (n = 3 cerebral organoids, error bars represent standard error of the mean). **d** Correlation between AEI to *ADAR* expression (top) and to *ADARB1* expression (bottom) for all 8 week cerebral organoid cell-types. **e** Aggregated UMAP plot from 4 H1 teratoma processed

through the single-cell RNA sequencing pipeline. **f** Box-and-whiskers plot showing the AEI values for all major teratoma cell types, with each data point calculated as a pseudo-bulk value (n = 4 WT teratomas, centerline of the box represents the mean, box represents 95% confidence interval, and error bar is standard error of the mean) Significance is tested using a unpaired two-tailed t-test (*$p \le 0.05$, **$p \le 0.01$, ***$p \le 0.001$, and ****$p \le 0.0001$; ns, not significant.) **g** Correlation between AEI value to *ADAR* expression (top) and to *ADARB1* expression (bottom) for all teratoma cell-types.

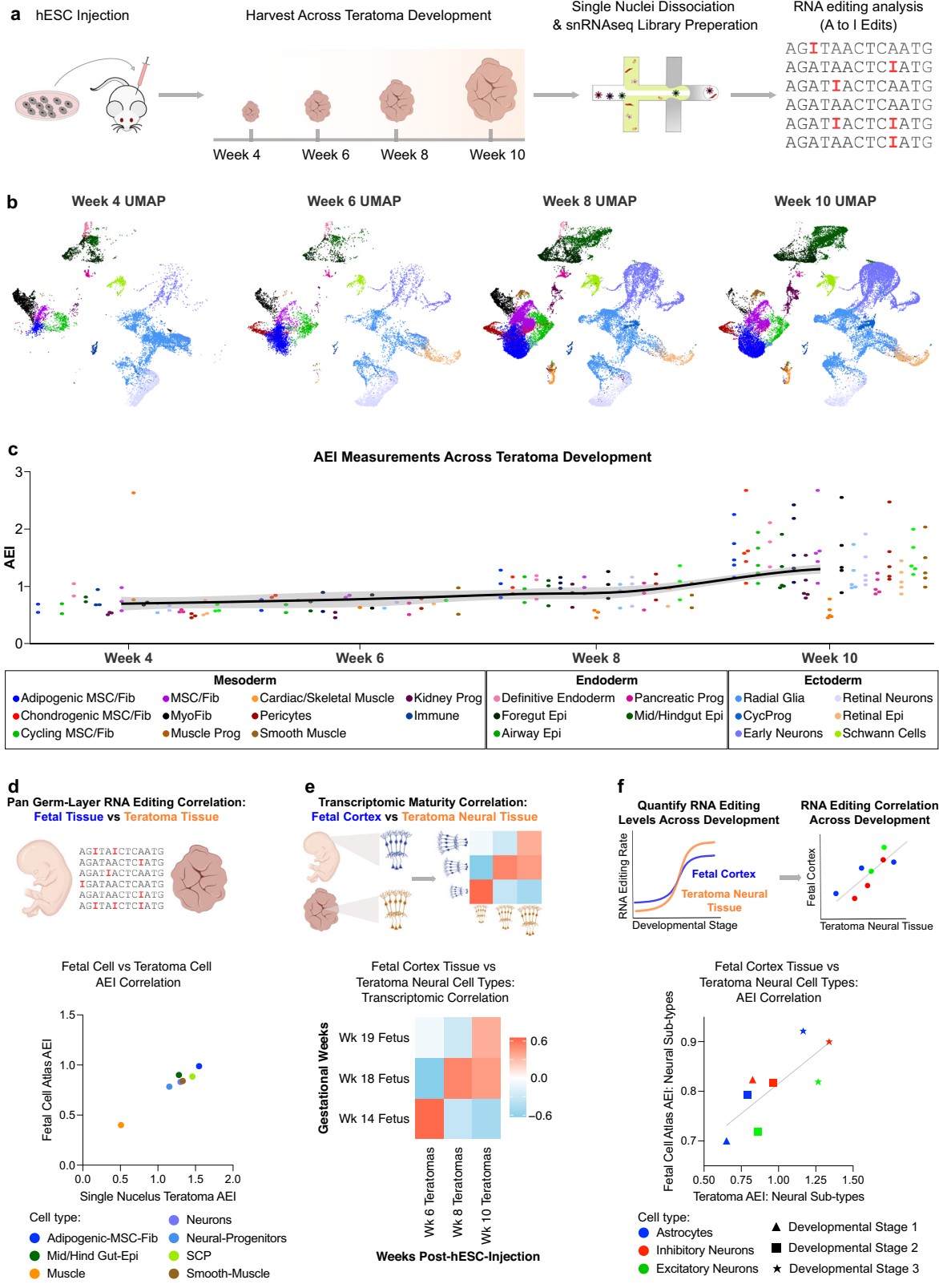

**Fig. 5 | Single nuclei RNA editing analysis across teratoma development.**
**a** Workflow schematic for running single nuclei RNA editing analysis across teratoma development. **b** Aggregated and downsampled UMAP plots from H1 teratomas across weeks 4–10 of development, processed through the single-cell RNA sequencing pipeline. 2–5 teratoma samples were processed for each time point. UMAP plots are downsampled so the total number of cells plotted is consistent across all time points. **c** AEI values for all teratoma cell-types across development. **d** Pan germ-layer correlation between teratoma cell-types and corresponding cell-types from the Human Fetal Cell Atlas database. **e** Transcriptomic correlation between time series teratomas and time-resolved human fetal cortex data between Gestational Weeks (GW) 14-19. **f** RNA editing rate, via AEI, correlation between time series teratomas and time-resolved human fetal cortex data between GW14–19.

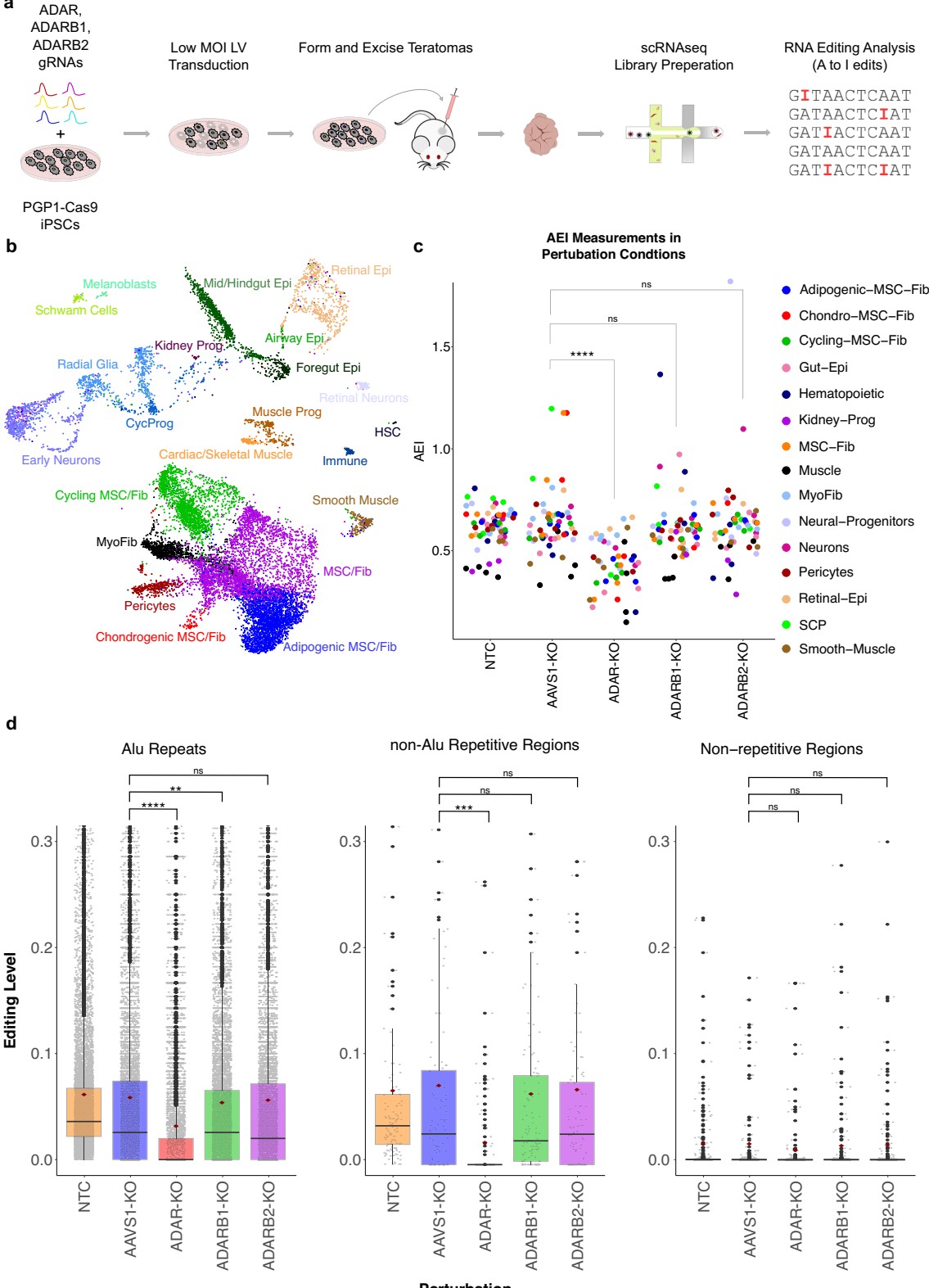

**Fig. 6 | RNA editing analysis in ADAR-KO teratomas. a** Workflow schematic for ADAR-KO screen in PGP1-Cas9 iPSCs, followed by teratoma formation and downstream single cell RNA editing analysis. **b** Aggregated UMAP plot from 4 PGP1-Cas9 + ADAR-KO teratomas. **c** AEI values for all teratoma cell-types in ADAR-KO teratomas. (*$p \leq 0.05$, **$p \leq 0.01$, ***$p \leq 0.001$, and ****$p \leq 0.0001$; ns, not significant). **d** Pan-teratoma site-specific RNA editing analysis across KO conditions. Centerline

of the box represents the mean, the box represents the 95% confidence interval, and the error bar is the standard error of the mean. (*$p \leq 0.05$, **$p \leq 0.01$, ***$p \leq 0.001$, and ****$p \leq 0.0001$; ns, not significant). n = 5166 ALU repeat sites, n = 84 non-ALU Repetitive Region sites, n = 338 non-Repetitive Region sites. ANOVA is used for comparing the mean and Tukey's significance test is used to determine which mean differences are statistically significant.

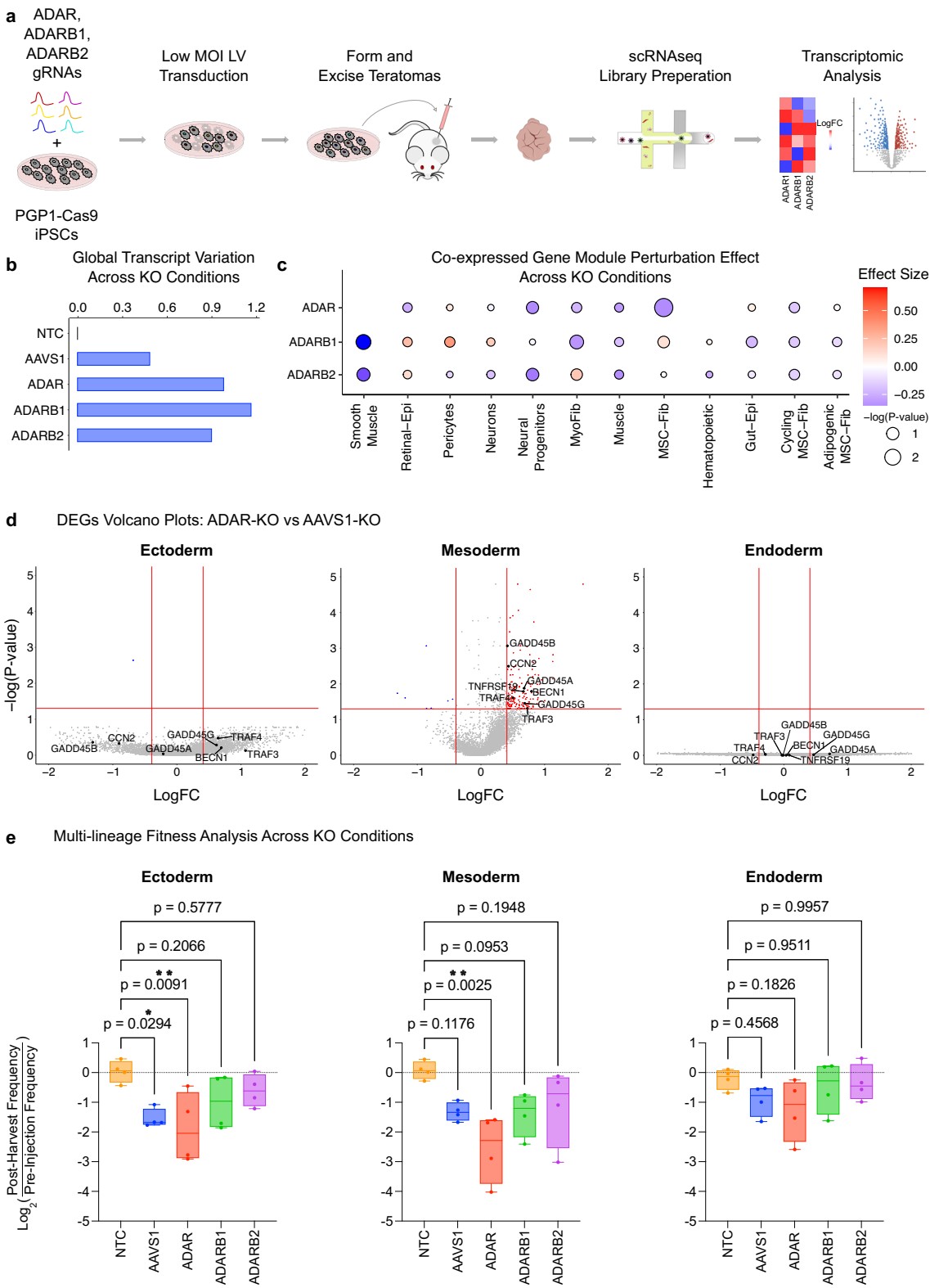

**Fig. 7 | Transcriptomic Analysis in ADAR-KO Teratomas Reveals Germ-Layer Specific Fitness Defects. a** Workflow schematic for ADAR-KO screen in PGP1-Cas9 iPSCs, followed by teratoma formation and downstream transcriptomic analysis. **b** Pan-teratoma global transcriptomic variation across perturbation conditions. **c** Cell-type specific gene-module effect across perturbation conditions. **d** Differential expressed genes (DEGs) for *ADAR*-KO versus *AAVS1*-KO across the three germ layers. Teratoma cell types from the same germ-layer are

transcriptomicaly grouped together. **e** Multi-lineage fitness analysis assessing the fitness defect across perturbation conditions. Cell counts from the same germ-layer are summed together. N = 4 unique teratomas used as independent biological replicates. Centerline of the box represents the mean, the box represents the 95% confidence interval, and the error bars are the standard error of the mean. (*$p \leq 0.05$, **$p \leq 0.01$, ***$p \leq 0.001$, and ****$p \leq 0.0001$; ns, not significant).

Interestingly, our analysis revealed that the contribution of *ADAR* to AEI variation in week 4 and 6 teratomas was less than 1%, but increased to approximately 10% by weeks 8 and 10 (Supplementary Fig. 6D). Furthermore, we observed a 2-fold increase in average AEI throughout teratoma development (pooled Cohen's d = 1.21) (Fig. 5c), which was largely driven by, increases in *ADAR* (pooled Cohen's d = 0.25) (Supplementary Fig. 6E). Notably, within Schwann Cell Progenitors (SCPs), *ADARB1* (p-value = 4.13E-02, Cohen's D = 1.73), in comparison to *ADAR* (p-value = 9.26E-02, Cohen's D = 1.21), seemed to mostly contribute to the significant rise in SCP AEI (p-value = 1.56E-03, Cohen's D = 2.01) throughout teratoma development (Supplementary Fig. 6E). These results motivated a deeper investigation into the reliability teratoma RNA editing dynamics.

To examine if the teratoma could serve as a reliable model for spatiotemporal RNA editing in human development, we correlated AEI values from teratomas with those from corresponding fetal cell types and found a strong correlation across cell-types from all germ-layers (Fig. 5D). Given the dynamic RNA editing profiles evident in fetal brain tissue (Fig. 2b, c)[27], our attention next shifted to teratoma neural tissue for a more in-depth analysis of time-resolved RNA editing in the teratoma.

Transcriptomic correlations across fetal cortex and teratoma neural cell-types revealed that neural cell-types within the teratoma progressively mature as the teratoma develops, as measured via the progressive increase in cosine similarity index with successive developmental time-points (Fig. 5e). This progressive transcriptomic maturation prompted us to compare RNA editing in teratoma neural tissue with that in fetal cerebrum tissue over time. Consistent with our maturity findings, the AEI measurements were correlated between teratoma and fetal samples, and showed a strong epitranscriptomic alignment throughout development (Fig. 5f).

In summary, our correlations of transcriptomic maturity and global editing with human fetal tissue corroborate the teratoma as an effective tool to model temporal RNA editing patterns during early fetal development. This motivated the use of teratomas as a platform to probe and interrogate the functions of ADARs in human development via pooled CRISPR-KO screens.

## RNA editing analysis in ADAR-KO teratomas

To directly assess the functions of ADAR family enzymes during cell-fate specification, we adopted a previously generated iPSC line, in which the Cas9 protein is knocked-in to the *AAVS1* locus of the PGP1 iPSCs (PGP1-Cas9s)[37]. PGP1-Cas9s were transduced with a gRNA library targeting *ADAR*, *ADARB1*, and *ADARB2*, injected into immunodeficient mice, and formed teratomas (ADAR-KO teratomas) (Fig. 6a). After a 10-week growth period, ADAR-KO teratomas were excised, and single-cell RNA sequencing libraries were generated. We confirmed that previously identified teratoma cell-types were present in ADAR-KO teratomas (Fig. 6b), and investigated RNA editing signatures across all cell-types utilizing both the AEI and a supervised site-specific analysis of A-to-I editing sites cataloged in existing RNA editing database[47].

In comparison to *AAVS1* double-strand break induction control cells (*AAVS1*-KO), we observed a statistically significant drop in AEI upon *ADAR* knockout (p-value = 9.6E-9, Cohen's d = -1.39) (Fig. 6c). In contrast, the disruption of *ADARB1* and *ADARB2* did not lead to discernible effects on global editing patterns. Furthermore, we examined site-specific RNA editing levels across bonafide editing sites cataloged in the RediPortal database[47] (Fig. 6d). Consistent with the above AEI analysis results, editing sites within *Alu* repeats (Cohen's d = -0.29) and non-*Alu* repetitive regions (Cohen's d = -0.45) exhibited a statistically significant drop in editing levels upon *ADAR* knockout (Fig. 6d). We also noticed a significant decrease in editing of *Alu* repeats following *ADARB1* knockout, although the impact is relatively minor compared to the effect of *ADAR* knockout (Cohen's d = -0.05) (Fig. 6d). In contrast, *ADARB2* knockout did not result in any significant changes in Alu

editing (Fig. 6d). These observations highlight the primary role of *ADAR* in A-to-I editing within *Alu* elements.

## Transcriptomic and fitness analysis in ADAR-KO teratomas

Motivated by the observed decline in RNA editing upon *ADAR*-KO, we extended our investigation to the transcriptomic level within ADAR-KO teratoma cell types (Fig. 7a). Employing the Earth's Mover Distance (EMD), a measure of dissimilarity between KO and non-targeting control (NTC) samples (refer Methods), we observed heightened global transcriptomic variation in *ADAR*-KO, *ADARB1*-KO, and *ADARB2*-KO samples, compared to *AAVS1*-KO samples (Fig. 7b).

To further investigate this transcriptomic variation upon *ADAR*-, *ADARB1*-, and *ADARB2*-KO, we conducted weighted correlation network analysis (refer Methods), revealing a reduced expression (negative effect size, Fig. 7c) of co-expressed genes across various cell types upon *ADAR* knockout, with the most pronounced impact observed in mesodermal cell-types, such as in the MyoFib, MSC/Fib, and muscle, as well as in ectodermal Neural-Progenitors (Fig. 7c). Next, we conducted differential expression analysis across knockout samples to examine these global transcriptomic impacts upon knockout.

First, DEGs were discovered at the germ-layer level across perturbations, and calculated relative to *AAVS1*-KOs (refer "Methods"). Discovered DEGs were used to conduct Gene Ontology (GO) term analysis in order to identify upregulated and downregulated gene families (refer Methods; Supplementary Figs. 7A–E). GO term analysis revealed the upregulation of the JNK pathway activation gene family in the Mesoderm *ADAR*-KO cells (Supplementary Fig. 7B). Interestingly, we observed that the JNK pathway genes were significantly upregulated solely in the mesoderm, and not in other germ layers (Fig. 7d). Considering the JNK pathway is a stress and apoptosis induction pathway[48], we next conducted a fitness analysis across perturbations. Initially, we observed a pan-teratoma fitness defect in *ADAR*-KO cells, while no significant fitness impact was observed in *ADARB1*-KO or *ADARB2*-KO cells (Supplementary Fig. 8A, B). Germ-layer level analysis unveiled a significant fitness defect in both the mesoderm *ADAR*-KO and ectoderm *ADAR*-KO (Fig. 7e), compared to NTC. However, the effect size of this fitness defect of *ADAR*-KO, compared to NTC, is most pronounced in the mesoderm (Cohen's d = 3.12) compared to the ectoderm (Cohen's d = 2.16). These results mirror the significant fitness defect—which ultimately leads to embryonic lethality—seen in *Adar*-KO mice[33], further corroborating the teratoma as a faithful model which recapitulates phenotypes seen in bonafide mammalian development.

## Cell-type enrichment in ADAR-KO teratomas suggests ADAR as an inhibitor of human adipogenesis

Motivated by the teratoma's unique multi-lineage nature and ability to faithfully recapitulate developmental phenotypes, we investigated the ADAR proteins' significance in human embryonic cell-fate specification, which remains insufficiently investigated and poorly understood. Leveraging ADAR-KO teratomas as an experimental model, our objective was to uncover the extent of the ADAR family of proteins' influence on cell-fate determination across all germ-layers (Fig. 8a).

Our investigation commenced with an enrichment analysis of ADAR-KO teratomas, unveiling a pan-teratoma reduction in *ADAR*-expressing cells (Supplementary Fig. 8A) and discernible variations in cell type compositions across perturbations (Supplementary Fig. 8C). Employing a prior developed pipeline to calculate cell-type compositions following a pooled in vivo CRISPR-Cas9 knockout screen (refer Methods)[49], we sought to further analyze these observed effects via cell-type enrichment analysis. Enrichment analysis unveiled a significant increase of Adipogenic-MSCs in *ADAR*-KO cells compared to *AAVS1*-KO controls (Fig. 8b). This finding was unexpected, as it is contrary to the previously observed fitness defect in mesoderm *ADAR*-KO cells (Fig. 7e). DEG analysis in *ADAR*-KO Adipogenic-MSCs vs AAVS1-KO Adipogenic-MSCs revealed many differentially expressed adipogenic, JNK/Stress

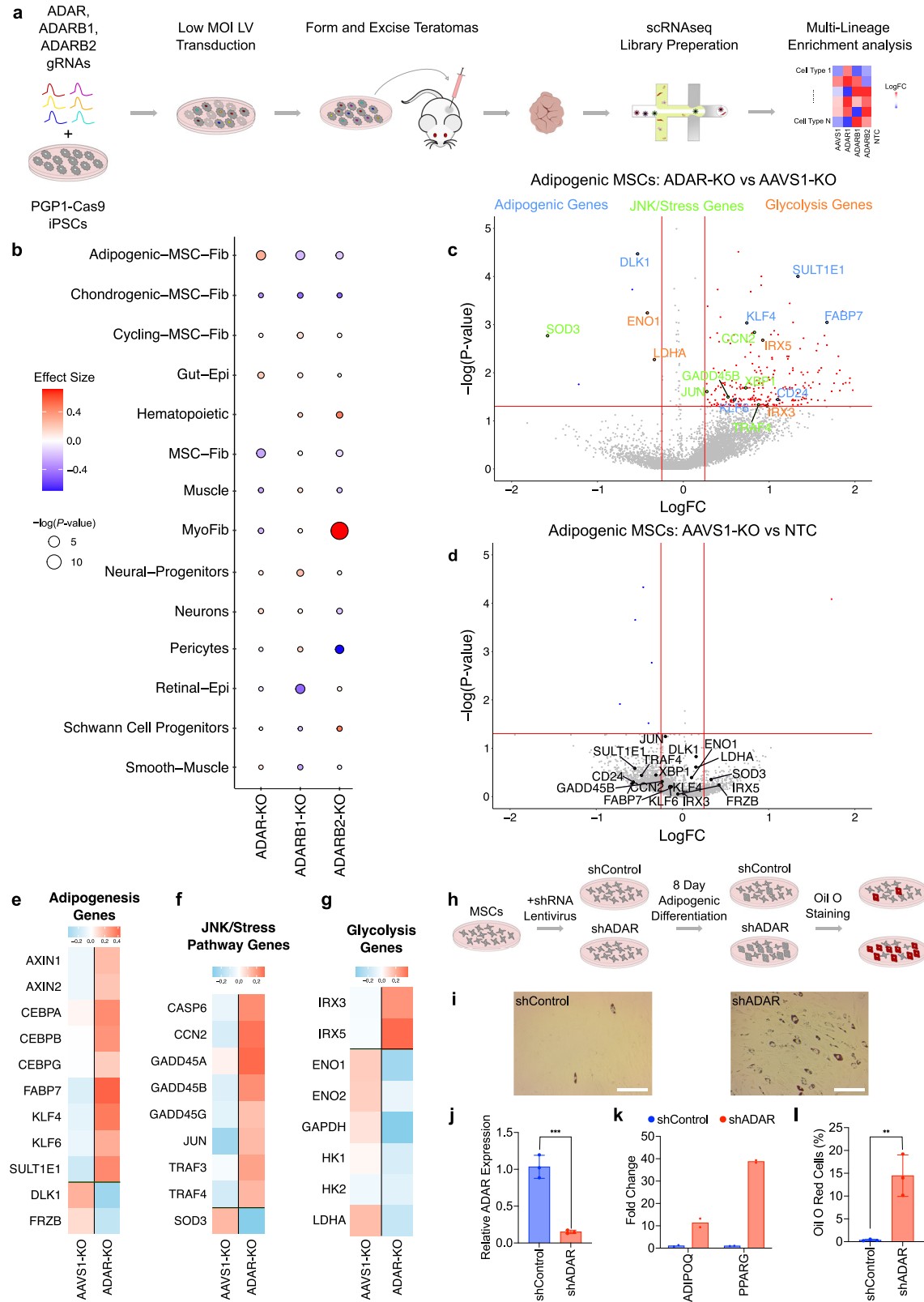

pathway, and glycolysis genes (Fig. 8c). In contrast, we found no significant DEGs when comparing AAVS1-KO Adipogenic-MSCs to NTC Adipogenic-MSC controls (Fig. 8d). Further analysis into this JNK-mediated stress response in *ADAR*-KO Adipogenic-MSCs revealed a strong stress response gene expression signature (Fig. 8e). In line with enrichment results (Fig. 8b), we correspondingly observed a strong pro-adipogenic gene expression signature (Fig. 8f).

Inspired by recent literature that links viral RNA infection, and the subsequent stress response, to downregulated glycolysis via MAVS signaling[50,51], we measured the expression of glycolysis-related genes to see if the *ADAR*-KO perturbation recapitulated this phenotype. We observed a decreased expression in many glycolytic genes (Fig. 8g). Due to a recent report demonstrating that IRX proteins trigger adipogenesis from hMSCs via repressed glycolysis[52], we measured IRX3

**Fig. 8 | Enrichment analysis in ADAR-KO teratomas reveals _ADAR_ as a potential inhibitor of human adipogenesis. a** Workflow schematic for ADAR-KO screen in PGP1-Cas9 iPSCs, followed by teratoma formation and downstream cell-type enrichment analysis. (B) Enrichment analysis depicting cell-type specific enrichment or depletion across perturbation conditions. **c** DEGs from comparing Adipogenic MSC _ADAR_-KO cells versus Adipogenic MSC AAVS1-KO cells. Genes related to adipogenesis (blue), JNK/stress pathways (green), and glycolysis (red) are highlighted. **d** DEG analysis comparing Adipogenic MSC _AAVS1_-KO cells versus Adipogenic MSC NTC cells. Genes related to adipogenesis, JNK/stress pathways, and glycolysis are denoted. **e** Gene expression analysis across a subset of adipogenic pathway genes in Adipogenic MSC _ADAR_-KO cells versus Adipogenic MSC AAVS1-KO cells. **f** Gene expression analysis across a subset of JNK/stress pathway genes in Adipogenic MSC _ADAR_-KO cells versus Adipogenic MSC _AAVS1_-KO cells. **g** Gene

expression analysis across a subset of glycolysis pathway genes in Adipogenic MSC _ADAR_-KO cells versus Adipogenic MSC _AAVS1_-KO cells. **h** Schematic depicting experimental validation of _ADAR_'s role in adipogenic differentiation from MSCs. **i** Oil O Red staining of 8 day differentiated Adipogenic-MSCs (Scale bar = 250 μm). **j** _ADAR_ transcript quantification in MSCs post shRNA transduction, at day 0 of adipogenic differentiation (n = 3 biological replicates, error bars represent standard error of the mean). (*$p \leq 0.05$, **$p \leq 0.01$, ***$p \leq 0.001$, and ****$p \leq 0.0001$; ns, not significant.) **k** _ADIPOQ_ and PPARG transcript quantification in MSCs post shRNA transduction, at day 8 of adipogenic differentiation (n = 2 biological replicates). **l** Quantification of Oil O red percent positive Adipogenic-MSCs. Full tilescan images of a 48-well are taken and quantified with respect to total DAPI+ cells (n = 3 biological replicates, error bars represent standard error of the mean). (*$p \leq 0.05$, **$p \leq 0.01$, ***$p \leq 0.001$, and ****$p \leq 0.0001$; ns, not significant).

and IRX5 expression levels and observed an increase in their expression levels (Fig. 8g).

To validate the observed increase in Adipogenic-MSCs within teratoma cells following _ADAR_-KO, we conducted experiments using primary human MSCs in vitro. These cells were transduced with lentivirus carrying _ADAR_-targeting shRNA (LV-shADAR) or a non-targeting shRNA control (LV-shControl) (Fig. 8h). We first verified the effectiveness of _ADAR_ knockdown after shADAR transduction at the onset of adipogenic differentiation (day 0) (Fig. 8j) Furthermore, we observed enhanced adipogenesis at day 8 of adipogenic differentiation in shADAR samples, as indicated by an upregulation of adipogenic markers _ADIPOQ_ and _PPARG_ (Fig. 8k) and a significant increase in Oil Red O positive percent cells compared to the shControl (Fig. 8i, l). These findings validate the observed adipogenic enrichment in _ADAR_-KO teratomas.

We further validated this finding using an iPSC-derived MSC differentiation model. Two distinct iPSC lines (KOLF2.1Js and PGP1s) were differentiated to MSCs and then similarly transduced with LV-shADAR, along with LV-shControl (Supplementary Fig. 9A). As seen in primary MSCs, following ADAR repression (Supplementary Fig. 9B) we observed increased adipogenic differentiation, as measured by Oil O Red positive staining (Supplementary Fig. 9C). These findings further substantiate the observed adipogenic enrichment in _ADAR_-KO teratomas.

We hypothesize that an increase in dsRNA following _ADAR_-KO[53] triggers a similar MAVS-mediated stress response following viral infection. Furthermore, we suggest that the stress induced by JNK pathway activation triggers adipogenesis, and synergistically interacts with the IRX-regulated reduction in glycolysis—a known driver of adipogenesis from hMSCs[52]. This intricate interplay potentially suggests putative mechanistic underpinnings behind the observed enrichment of Adipogenic-MSCs following _ADAR_ knockdown.

## Discussion

Beyond their known role in editing dsRNA to prevent unwanted innate immune responses, the diverse functions of ADAR proteins, especially in embryonic development, are understudied. Here, we characterized spatiotemporal RNA editing patterns across various human fetal tissues, extended our investigations to lab-grown developmental model systems, and probed the functional roles of ADARs in cell-fate determination across all three germ layers via CRISPR-Cas9 KO screening in the teratoma.

Our time-course analysis of human fetal-to-adult organ data identified trends that motivate future investigations into the spatiotemporal role of ADARs beyond brain tissue[27]. For instance, we noted sharp drops in global editing levels during early gestation (4–9 wpc) in both heart and liver tissue. Investigating whether this conserved phenomena is coincidental or driven by convergent underlying mechanisms will be an intriguing avenue for future analysis. Furthermore, the sharp decline in AEI and _ADAR_ expression between years 25 and 63 of

testis development suggests a senescence- or aging-related phenotype, as diminished _ADAR_ expression is associated with senescence[54]. This observation raises the possibility that the testis may exhibit an accelerated-aging phenotype, in relation to the other organs we've studied, given that the testis are the only examined organ tissue in our analysis which exhibits a decrease in _ADAR_ expression in the later time-points. Considering our dataset encompasses individuals up to 63 years old, it would be intriguing to explore time-series transcriptomic profiles in more advanced stages of aging (beyond 63 years of age) to determine whether this decline in _ADAR_ expression is a convergent trend across other organ tissues as well.

Prenatal RNA editing dynamics particularly intrigued us, as they may offer glimpses into the elusive black-box that is mammalian development. Towards this, we characterized RNA editing trends in cerebral organoids and teratomas, as both provide a lab-grown source of prenatal-to-perinatal tissue that can be experimentally manipulated. We observed that the teratoma is a powerful model system as it allows the exploration of RNA editing profiles across all three germ-layers simultaneously, and faithfully recapitulates RNA editing trends seen in human fetal development. Lastly, through our unbiased, pan-tissue screening in the teratoma, we observed a fitness defect in _ADAR_-KO cells, that is most strongly seen in mesodermal cells, and uncovered an _ADAR_-mediated role in inhibiting adipogenesis from human MSCs.

_ADAR_'s role in murine cell adipogenesis has been appreciated in recent years[55,56], but has yet to be explored in a human context. We propose this adipogenic effect is in part driven by a decrease in _ADAR_ expression, putatively leading to an increase in cellular dsRNA. The presence of dsRNA triggers a stress response[57], repressing glycolysis[50,51] and potentially driving adipogenic programs downstream[52]. Lastly, although there is substantial research linking inflammation and obesity[58,59], the link to decreased RNA editing has yet to be explored. Our findings indicate that inflammation triggered via _ADAR_ downregulation, potentially from cellular senescence and physiological aging[54], may be a driver for obesity. Future studies could compare RNA editing levels in adipogenic tissue from obese and non-obese individuals to investigate if these trends hold in an adult organism.

Our findings further corroborate the importance of ADARs in human development, and extend previous spatiotemporal editing investigations. We also demonstrated that the teratoma could be used to model prenatal RNA editing dynamics, and uncovered the role of ADARs in mesenchymal stem cell fate specification. The link between _ADAR_ dysregulation and adipogenesis could also motivate the development of therapeutics for obesity.

A limitation of this work is the use of 3' biased RNA sequencing libraries, which limited the scope of the examined editome. This leads to a loss of editing information across the RNA transcript, most notably in re-coding regions. Future studies could involve generating single-cell full-length RNA sequencing libraries, which will enable the tracking of re-coding regions, at the single-cell level.

The epitranscriptome is a powerful regulatory system within the cell that is understudied and holds great potential for discoveries in gene expression dynamics. This study is an extensive investigation of RNA editing and ADAR expression dynamics across various human tissues, providing a unique opportunity to uncover RNA editing trends and the diverse functional roles of ADAR proteins in embryonic development and organogenesis.

## Methods

### AEI RNA editing analysis
Raw sequencing reads are parsed to create pseudobulk reads for each cell state, such as cell types, batches of experiment and knockout conditions and the *dedup* function from umitools package is used to remove PCR duplicate reads. *Alu* editing index (AEI) was calculated for deduplicated bam files for each experiment condition using the RNAEditingIndexer v1.0. AEI is an aggregated index representing the ratio of edited reads (A-to-G mismatches) over the total coverage of adenosine bases in *Alu* elements and has been demonstrated as a robust index with respect to coverage, read lengths as well as batch effects introduced by sample preparation. *Alu* repeats outlined by UCSC genome browser were supplied to designate regions where AEI was calculated over and SNP sites included in hg38CommonGenomicSNPs150 were discarded.

### Site-specific RNA editing analysis
Editing levels at known sites were calculated similar to existing computational pipelines[27]. First, coordinate-sorted BAM files of mapped, parsed and deduplicated reads were generated with STAR. Then, nucleotide coordinates of known editing sites previously identified in REDIportal were examined with customized scripts with samtools mpileup function[47]. Filters were applied to remove SNP sites in common genomic variation in dbSNP (v150) and sites within 5 base pairs to read ends and splice sites. At least 5 reads covering the site and at least 3 edited reads observed were required to call an editing event for single cell/single nucleus sequencing read pseudobulk samples.

### Differential RNA editing analysis and gene ontology analysis
We adopted a similar approach to existing computational pipelines to catalog common A-to-I editing sites and identify differentially edited sites across human development[27]. To ensure capturing of common editing sites across samples, we imposed stringent criteria for high-quality sites, keeping sites that have at least 60% detection rate and at least 5% mean editing rate in samples subsetted from each organ. Additionally, samples with more than 20% missing values were excluded. Confirming that the processed RNA editing matrices exhibited less than 7% missing data on average, the mice R package[60] is employed to impute the missing values using the predictive mean matching method with five multiple imputations and 30 iterations.

To uncover differentially edited sites, we utilized the limma R package[61]. Potential confounders including sex, expression level of ADAR and ADARB1 (both enzymes demonstrate importance in regulating development and varying abundance) were taken into account as covariates for linear modeling. We adjusted for multiple testing using the Benjamini-Hochberg procedure and sites with FDR corrected p-values < 0.05 were deemed as significant. Genes containing A-to-I editing sites with significant levels of editing changes were recognized as differentially edited genes.

Differentially edited genes were functionally annotated using the ToppGene Suite[62].

### RNA editing analysis in bulk human organs
RNA sequencing data for all bulk human organs was retrieved on Array Express via accession code E-MTAB-6814[38]. Time series data was split into 4 groups, based on their developmental stage annotations: Group 1) 4 weeks post conception (wpc) to 10 wpc were binned into "early gestation"; Group 2) 11 wpc to 20 wpc were binned into "late gestation"; Group 3) 0 to 20 years of age were binned into "newborn-teeager"; and Group 4) 25-63 years of age were binned into "adult-senior". Comparisons were made between sequential time series groups—Group 1 versus 2, Group 2 versus 3, and Group 4 versus 4—in order to quantify the shift in RNA editing and ADAR expression between temporal stage transitions. A two-tailed Student's t-test was done to determine significance between groups and the Cohen's d test comparison was run to determine effect size.

AEI measurements were performed as mentioned above.

### RNA editing analysis in wild type teratomas
RNA sequencing data for teratomas was retrieved via NCBI GEO accession code GSE: 156170. AEI measurements were performed as mentioned above.

### RNA editing analysis in human fetal organs
RNA sequencing data for fetal organs was retrieved via raw data provided at dbGaP (accession number phs002003.v1.p1). AEI measurements were performed as mentioned above.

### Figure generation
All figures and schematics were generated using Inkscape is a free graphic and design editor.

### Reporting summary
Further information on research design is available in the Nature Portfolio Reporting Summary linked to this article.

## Data availability
The raw and processed data generated from this study are available at Gene Expression Omnibus with accession code GSE248941. Source data are provided with this paper.

## Code availability
Customized scripts used for analysis are available at this github repository: SammiLyu/scScreens_ADARs [https://github.com/SammiLyu/scScreens_ADARs].

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

## Acknowledgements

The authors would like to thank Kian Kalhor, Yan Wu, and Ramin Dailamy for useful discussions and support throughout all stages of the study. This work was generously supported by UCSD Institutional Funds, NIH grants (R01HG012351, OT2OD032742, R01NS131560, U54CA274502), and a Department of Defense Grant (W81XWH-22-1-0401). This publication includes data generated at the UC San Diego IGM Genomics Center utilizing an Illumina NovaSeq 6000 that was purchased with funding from a National Institutes of Health SIG grant (#S10 OD026929).

## Author contributions

Conceptualization and design: P.M., A.D., W.L., A.M.; Experiments: A.D., W.L., J.R., S.N., D.M., R.P.; Computational analyses: W.L., M.T., M.B., K.Z.; Writing: P.M., A.D., W.L. with input from all authors.

## Competing interests

P.M. is a scientific co-founder of Shape Therapeutics, Boundless Biosciences, Navega Therapeutics, Pi Bio, and Engine Biosciences. A.M. is the co-founder of and has an equity interest in TISMOO, a company dedicated to genetic analysis and human brain organogenesis, focusing on therapeutic applications customized to autism spectrum disorders and other neurological diseases. K.Z. is a full-time employee and equity holder of the Altos Labs. The remaining authors declare no competing interests. The terms of these arrangements have been reviewed and approved by the University of California, San Diego in accordance with its conflict of interest policies.
