## [Transparent Peer Review file · Nature Communications]

Charting and Probing the Activity of ADARs in Human Development and Cell-Fate Specification

Corresponding Author: Dr Prashant Mali

Version 0:

Reviewer comments:

Reviewer #1

(Remarks to the Author)

In this manuscript, Dailamy and collaborators address A-to-I editing dynamics across various stages of human organ development, spanning from fetal to adult stages. Employing a comprehensive approach, they utilize *in vitro* models such as brain organoids and *in vivo* models like teratomas, both derived from the differentiation of human embryonic stem cells (ESCs), to unravel the patterns of A-to-I editing during human development. The investigation extends further with a focus on CRISPR knockout (CRISPR-KO) and single-cell RNA sequencing (scRNA-seq) techniques applied to teratomas, to elucidate the roles of ADAR1, ADAR2, and ADAR3. Notably, the study highlights the involvement of ADAR1 in mesenchymal differentiation.

While the study introduces novel insights into the A-to-I dynamics within models of human development and differentiation, it predominantly adopts a descriptive approach, providing limited exploration into the significance of A-to-I editing in both *in vivo* and *in vitro* development contexts. Additionally, the A-to-I editing analysis is confined to Alu elements, omitting crucial details regarding (1) the identity of mRNAs harboring these elements in their introns and/or UTRs and (2) the functional implications of editing in these specific RNA species. Last, it is worth noting that previous research has already addressed the role of ADAR1 in adipogenesis (e.g., doi: 10.1242/jcs.259333) and human brain development (e.g., doi.org/10.1016/j.celrep.2022.111585).

Below is a description of additional main concerns:

1. The authors show differential Alu editing index dynamics in several human organs from early gestation until adult-senior age. In order to gain a deeper insight into the meaning of such editing differences, it would be advisable to identify the mRNAs bearing these Alus, either using the same or a different A-to-I mapping pipeline. The authors should also address whether the change in A-to-I editing is correlated with a change in expression of the edited targets.
2. Given the immunogenic potential of endogenous RNA double-stranded structures, A-to-I editing of dsRNA is crucial to prevent activation of the innate immune response. The authors should explore whether changes in A-to-I editing during the fetal-to-adult transition correlate with pathways or processes related to innate immunity.
3. While Alus are primarily targeted by ADAR1, with ADAR2 playing a more significant role in recoding A-to-I events in mRNAs, the authors suggest that ADAR2 drives Alu editing during hindbrain development. Could the authors explain or speculate on this apparent contradiction?
4. In Fig. S4E, the authors indicate that in Schwann Cell Progenitors (SCPs) ADAR2 contribution is higher than ADAR1 in the significant rise of AEI throughout teratoma development. However, the image does not clearly show appreciable differences between ADAR1 and ADAR2 Cohen's d values. If the authors wish to emphasize this result, an additional panel should be included with a more accurate representation, making the differences clearer to the reader.
5. The authors evaluate fitness in their models following ADAR CRISPR knock-outs. A more detailed explanation of how fitness is determined would enhance the clarity of this analysis.
6. While the authors primarily employed single-cell RNA-seq (scRNA-seq), they switched to single-nuclei RNA-seq (snRNA-seq) in samples across teratoma development (Figure 4). The rationale for this choice should be explained. Additionally, considering that many Alus are located in introns, which are present in nuclear RNA but absent in mature cytoplasmic mRNAs, the authors should address whether this analysis introduces any bias when comparing to other tissues.

Minor comments:

7. The nomenclature for ADAR1, ADARB1, and ADARB2 is inconsistent, mixing non-official (i.e., ADAR1) and official (i.e., ADARB1 and ADARB2) protein names. It is recommended to use either "ADAR1, ADAR2, and ADAR3" or "ADAR, ADARB1, and ADARB2" consistently.
8. Several panels are not referenced in the main text (i.e., Fig. 2H, Fig. 3D, Fig. S1C, and Fig. 4B) and should be appropriately cited.
9. In the introduction section, the statement that "A-to-I RNA editing leads to the incorporation of inosines" should be revised to accurately reflect that A-to-I editing results in the deamination of adenosines, converting them to inosines.
10. In bulk human organs, the number of datasets (n) for each organ should be indicated.

Reviewer #2

(Remarks to the Author)

The manuscript by Dailamy et al. titled "Charting and Probing the Activity of ADARs in Human Development and Cell-Fate Specification" presents a comprehensive study on the role of Adenosine Deaminases acting on RNA (ADARs) in human development, focusing on their functions in early cell fate specification. The research is well-structured and contributes to the field of developmental biology and gene regulation. The main strengths:

- 1) The multi-pronged methodology combining time-course RNA editing profiles in human organs and hPSC differentiation models is commendable. This approach provides a holistic view of ADARs' roles across developmental stages.
- 2) The utilization of both brain organoids and teratomas for experimental probing of ADARs is technically sound. This dual-model approach enhances the validity of the findings.
- 3) The thorough analysis of RNA editing levels, ADAR enzyme expression dynamics, and single-cell RNA editing profiles is well executed. The use of advanced analytical methods, such as single-cell CRISPR-KO screens, adds robustness to the findings.
- 4) The discovery of ADAR1's role in mesenchymal differentiation and its potential implication in obesity-related phenotypes is a significant contribution to the field.

Conclusion: This manuscript sheds new light on understanding the role of ADARs in human development. The comprehensive approach, rigorous data analysis, and novel insights pave the way for future research in this area. However, some areas require further improvement before the manuscript can be accepted for publication:

- 1) While the study focuses on the developmental aspect and provides a discussion on the implications of ADARs in certain tissue/organ development and diseases, not a single edited target gene was mentioned (discovered) to offer deeper insights into the molecular players downstream ADARs whose editing defects may contracture to the observed/speculated functional readouts. Additional work is needed to put ADARs and their edited target genes into the context of the underlying biology.
- 2) The interpretation of the data, especially in complex systems like human development, is crucial. Ensuring that the conclusions drawn are supported by the data and validated by independent methods or data sets is essential for robustness. The findings from human teratoma and organoids should be validated in the human PSC differentiation model along specific lineages, even better and most relevant, using human synthetic embryos such as hPSC-derived blastoids.

Version 1:

Reviewer comments:

Reviewer #1

(Remarks to the Author)

The authors have addressed all the points raised by both reviewers. The manuscript now includes additional data with novel conclusions on tissue-specific and pathway-enriched edited targets which improves the manuscript, in the opinion of the reviewer.

Reviewer #2

(Remarks to the Author)

The authors have done a commendable job of revising and improving the manuscript. The study now presents some interesting findings and resourceful datasets that will benefit the scientific community. The manuscript can be accepted for publication with the remaining minor points addressed.

1. Page 4: Fig. 2F, 2H: the AEI upregulation was mentioned to coincide with ADARB1 downregulation in the Kidney. But, how AEI got upregulated was not mentioned or discussed. Did the authors check other ADAR members?
2. Page 6: "Innate Immunity" GO term was mentioned in the result text for Fig. 3C; however, Fig. 3C did not present the "innate Immunity" GO term.
3. Page 6: AEI (blue line) should be pointed out in the results text before (Fig. 3E-J) to be consistent with the references to the other two colored lines mentioned already.
4. Page 7: "Supplementary Fig. 3" was referred to with a single summary statement without any meaningful explanation or mention of all the 10 (A-J) figure panels. What those 10 figure panels are should be explained.
5. Page 9: "cosine similarity index" was mentioned and referred to (Fig. 5F), which was not present in that Fig. 5F panel.

RESPONSE TO REVIEWER COMMENTS

We thank the reviewers for their efforts on our manuscript. Overall, we found the critiques to be very constructive and helpful, and have strived during this review period to address every single aspect that was raised, resulting in additional new analyses and new experiments, including across different iPSC lines. This has helped strengthen our conclusions and improve the overall rigor of our study. Below we provide our point-by-point response:

Reviewer #1 (Remarks to the Author):

In this manuscript, Dailamy and collaborators address A-to-I editing dynamics across various stages of human organ development, spanning from fetal to adult stages. Employing a comprehensive approach, they utilize in vitro models such as brain organoids and in vivo models like teratomas, both derived from the differentiation of human embryonic stem cells (ESCs), to unravel the patterns of A-to-I editing during human development. The investigation extends further with a focus on CRISPR knockout (CRISPR-KO) and single-cell RNA sequencing (scRNA-seq) techniques applied to teratomas, to elucidate the roles of ADAR1, ADAR2, and ADAR3. Notably, the study highlights the involvement of ADAR1 in mesenchymal differentiation.

While the study introduces novel insights into the A-to-I dynamics within models of human development and differentiation, it predominantly adopts a descriptive approach, providing limited exploration into the significance of A-to-I editing in both in vivo and in vitro development contexts. Additionally, the A-to-I editing analysis is confined to Alu elements, omitting crucial details regarding (1) the identity of mRNAs harboring these elements in their introns and/or UTRs and (2) the functional implications of editing in these specific RNA species. Last, it is worth noting that previous research has already addressed the role of ADAR1 in adipogenesis (e.g., doi: 10.1242/jcs.259333) and human brain development (e.g., doi.org/10.1016/j.celrep.2022.111585).

Firstly, in response to the reviewer's observation regarding previous studies on ADAR's involvement in adipogenesis, the cited 2022 report focuses on ADAR's role in murine adipogenesis. Our study, however, utilizes an unbiased human pluripotent stem cell differentiation fitness screen, revealing adipogenic enrichment. Notably, this is the first instance of reporting ADAR's role in human cell adipogenesis. Given the acknowledged species-specific differences between human and murine adipogenesis in existing literature (10.3389/fcell.2022.1003118 ;

10.1016/j.cmet.2013.05.020 ; 10.3389/fcvm.2017.00027 ; 10.1128/MCB.01147-13), our findings bear significance for understanding human adipogenesis biology.

Furthermore, regarding the reviewer's mention of prior research on ADARs in human brain development, we acknowledge the existing body of literature on this topic. In our manuscript, we conducted RNA editing analysis on various tissues, including brain, heart, liver, kidney, and testis. Our focus on brain tissue RNA editing primarily serves to validate the integrity of our analyses and ensure alignment with established peer-reviewed literature on RNA editing.

Below is a description of additional main concerns:

1. The authors show differential Alu editing index dynamics in several human organs from early gestation until adult-senior age. In order to gain a deeper insight into the meaning of such editing differences, it would be advisable to identify the mRNAs bearing these Alus, either using the same or a different A-to-I mapping pipeline. The authors should also address whether the change in A-to-I editing is correlated with a change in expression of the edited targets.

We addressed this crucial point by employing a site-specific RNA editing pipeline. This pipeline identified edited mRNA sites within the REDportal table across all bulk organ time-series data (forebrain, hindbrain, liver, heart, kidney, and testis). Subsequently, we examined differential editing levels in these sites between postnatal and prenatal samples using customized R scripts, akin to established computational methodologies (PMID: 36323256). The resulting delta editing values (postnatal vs prenatal) were then correlated with the log fold change in expression levels of corresponding genes between postnatal and prenatal samples (**Response Figure 1**).

Response Figure 1: Schematic depicting the site-specific RNA editing analysis pipeline for developmental bulk organ samples. After bulk organ samples were grouped into prenatal and postnatal groups, differential editing analysis was conducted between these two developmental

groups. Finally, prenatal versus postnatal differences in RNA editing levels (delta editing rate) were correlated with corresponding gene expression level changes (log fold-change).

After identifying RNA editing sites at a site-specific level, we next classified editing sites into their corresponding genetic regions, and found that the majority of identified sites are located in 3'UTRs (**Response Figure 2A**). Furthermore, we found that the largest number of editing sites were identified in the hindbrain (**Response Figure 2A**).

Next, we identified sites that are differentially edited between prenatal and postnatal samples across all tissues (**Response Figure 2B**). The delta editing rate of these differentially edited sites were then correlated with the log fold change (logFC) of the corresponding gene expression level changes (**Response Figure 2C**). Across all organs, we observed a slightly negative correlation ($R = -0.133$) between the change in editing levels and gene expression across all identified differentially edited genes (**Response Figure 2C**).

Response Figure 2: (A) Total number of RNA editing sites discovered across all examined organs. **(B)** Total number of prenatal versus postnatal differentially edited RNA editing sites discovered across all examined organs (n.d = none detected). **(C)** Prenatal versus postnatal differences in RNA editing levels (delta editing rate/gene) were correlated with corresponding gene expression level changes (log fold-change/gene) for all genes captured across all organs.

Delta editing rates per gene were averaged across all delta values for each site per unique gene. Correlation was measured using a Pearson's Correlation Coefficient and 95% confidence intervals are plotted for each genetic region.

Intrigued by this slightly negative correlation, we then separated the data at the organ level, and mostly observed weak trends across the forebrain, hindbrain, kidney and testis tissues (**Response Figure 3A-D**). Interestingly, we noticed a slightly more convincing negative correlation in liver tissues (**Response Figure 3E**), and when separating out the sites at a genetic region level, we noticed a strong negative ($R = -0.544$) correlation in liver intronic editing sites (**Response Figure 3F**). This negative correlation prompted a functional characterization of these genes. Using the ToppGene Suite, we observed GO terms associated with cholesterol biosynthesis and metabolism (**Response Figure 3G**). The input genes that are associated with these cholesterol GO terms are SREBF1, HMGCS1, FDPS, FADS2, and EBP, and to our knowledge there are no literature that explores the effects and dynamics of RNA editing in these cholesterol genes during the fetal-to-adult transition. A further exploration into the significance of RNA editing at these sites may potentially shed light on novel cholesterol gene regulation modalities.

Response Figure 3: (A-F) Prenatal versus postnatal differences in RNA editing levels (delta editing rate/gene) were correlated with corresponding gene expression level changes (log fold-change/gene) for all sites captured in the forebrain (A), hindbrain (B), kidney (C), testis (D), liver (E), and liver intronic sites only (F). (G) GO term functional annotation of differentially edited liver intronic sites.

2. Given the immunogenic potential of endogenous RNA double-stranded structures, A-to-I editing of dsRNA is crucial to prevent activation of the innate immune response. The authors should explore whether changes in A-to-I editing during the fetal-to-adult transition correlate with pathways or processes related to innate immunity.

Towards addressing the reviewer's question, we delved into the RNA editing sites identified through the site-specific RNA analysis pipeline on the time-series bulk organ data. After compiling a list of differentially edited sites between prenatal and postnatal stages across all tissues, we investigated the convergence of these sites across organs (Response Figure 4A). This

exploration aimed to uncover genes with differentially edited sites common to multiple tissue types, shedding light on potential developmentally conserved RNA editing mechanisms.

We observed 58 prenatal versus postnatal differentially edited sites that were shared across all organ datasets (**Response Figure 4B**). Intrigued by this convergence, we conducted functional annotation on these genes using the ToppGene Suite and identified GO terms associated with innate immunity and DNA replication (**Response Figure 4C**). Notably, EIF2AK2 (also known as PKR) and MAVS, involved in viral response and innate immunity pathways, were found to have differentially edited sites in all analyzed tissues (**Response Figure 4D**). Thus, through an unbiased look at the convergence of prenatal versus postnatal differentially edited RNA editing sites across organs from all three germ-layers, we identified that there is a significant shift in RNA editing of innate immunity related genes during the fetal-to-adult transition.

Response Figure 4: (A) Schematic depicting the site-specific RNA editing analysis pipeline for developmental bulk organ samples. After bulk organ samples were grouped into prenatal and postnatal groups, differential editing analysis was conducted between these two developmental groups. Finally, a convergence analysis was conducted to find shared and unique differentially edited sites between organs. (B) Venn diagram illustrating the number of shared and unique prenatal versus postnatal differentially edited sites across all examined organ datasets. (C) GO

term functional annotation of the pan-organ shared differentially edited sites **(D)** List of the input genes that are associated with these corresponding convergent GO terms.

To assess how shifts in RNA editing correlate with gene expression, we utilized Gene Set Variation Analysis (GSVA) (PMID: 23323831), which condenses information from gene expression profiles to calculate gene scores for specific biological pathways. Specifically, we computed gene set scores for C2 gene sets from the Human MSigDB Collections (PMIDs: 21546393; 16199517), tracking them over time for the Viral Infection Gene Set (red line) and DNA Replication Gene Set (green line) **(Response Figure 5A-F)**. The global editing level of these samples, measured by the Alu Editing Index (AEI), was overlaid onto the graphs to visualize changes in global editing levels across development for each organ **(Response Figure 5A-F)**. We observed significant inverse correlations between the AEI and gene scores from these two pathways in the forebrain, hindbrain, and liver **(Response Figure 5A-F)**. This analysis underscores the crucial role of A-to-I editing in regulating gene expression in innate immunity and DNA replication pathways during the fetal-to-adult transition.

Response Figure 5: (A-F) Charting global A-to-I editing dynamics (AEI), along with the Viral Infection and DNA replication pathway gene scores, across developmental time, for the forebrain (A), hindbrain (B), heart (C), liver (D), kidney (E), and testis (F). Correlation between the gene score sets and the AEI were measured using the Pearson correlation coefficient.

3. While Alus are primarily targeted by ADAR1, with ADAR2 playing a more significant role in recoding A-to-I events in mRNAs, the authors suggest that ADAR2 drives Alu editing during hindbrain development. Could the authors explain or speculate on this apparent contradiction?

Although *ADARB1* is widely known for its role in mRNA re-coding events, it is also known to edit ALUs, thus we do not believe this observation is a contradiction. We observe a large increase in *ADARB1* expression during hindbrain development while *ADAR* expression stays relatively

unchanged, along with an increase in the Alu editing index. This observation is also reflected in Figure 3 of the following manuscript: [10.1038/s41592-019-0610-9](https://doi.org/10.1038/s41592-019-0610-9) , in which tissues with high *ADARB1* expression (arteries and brain) have the highest AEI levels.

4. In Fig. S4E, the authors indicate that in Schwann Cell Progenitors (SCPs) ADAR2 contribution is higher than ADAR1 in the significant rise of AEI throughout teratoma development. However, the image does not clearly show appreciable differences between ADAR1 and ADARB1 Cohen's d values. If the authors wish to emphasize this result, an additional panel should be included with a more accurate representation, making the differences clearer to the reader.

Via Cohen's d analysis on early (4-6 week) versus late (8-10 week) teratomas, we discovered that the effect size in the change of AEI was 2.0. We then measured expression changes in *ADAR* and *ADARB1* across early versus late teratomas, and noticed the effect sizes were 1.21 and 1.73, respectively. Thus, we suggest that *ADARB1*'s contribution to the increase in AEI is higher than that of *ADAR*, since the Cohen's d effect size value is greater ($1.73 > 1.21$). We can denote these differences in the figure by directly reporting the numbers in the graph.

5. The authors evaluate fitness in their models following ADAR CRISPR knock-outs. A more detailed explanation of how fitness is determined would enhance the clarity of this analysis.

For germ-layer level fitness analysis, the pre-injection frequency values for each unique guide were quantified from pre-injection ADAR-KO cell gDNA counts (pre-injection frequency). After teratoma harvest, cell counts for each unique guide were added up for each germ layer, and then divided by the total number of cells within that germ layer. This frequency (post-harvest frequency) was divided by the pre-injection guide frequency of each unique guide, and then transformed with log₂-scaling, resulting in the log₂(Fold-Change) value. Within each teratoma, log₂(Fold-Change) values for unique guides targeting the same gene were averaged, and n=4 log₂(Fold-Change) values were plotted using 4 unique ADAR-KO teratomas.

Cell counts from Adipogenic MSCs, Chondrogenic MSCs, Cycling MSCs, Hematopoietic Cells, Kidney Progenitors, MSC-Fibroblasts, Muscle, Myo-Fibroblasts, Pericytes, and Smooth Muscle Cells were summed to calculate the total number of mesodermal cells.

Cell counts from Neural-Progenitors, Neurons, Retinal-Epithelium, and Schwann Cell Progenitors were summed to calculate the total number of ectodermal cells.

Cell counts from the Gut-Epithelium and Pancreatic Progenitors were summed to calculate the total number of endodermal cells.

For pan-teratoma level fitness analysis, the calculations are run as described above except that all cells for each unique guide are summed and then divided by the total number of harvested cells to generate the post-harvest frequency, rather than splitting up cells by germ-layer.

This description was also added to the methods section of the manuscript, under the subsection titled “Germ-Layer and Pan-Teratoma Fitness Analysis”.

6. While the authors primarily employed single-cell RNA-seq (scRNA-seq), they switched to single-nuclei RNA-seq (snRNA-seq) in samples across teratoma development (Figure 4). The rationale for this choice should be explained. Additionally, considering that many Alus are located in introns, which are present in nuclear RNA but absent in mature cytoplasmic mRNAs, the authors should address whether this analysis introduces any bias when comparing to other tissues.

Given that many banked tissue samples are stored as fresh frozen OCT blocks, we sought a library preparation method that would be broadly applicable and compatible with this storage modality. Towards this, we explored single-nucleus RNA sequencing (snRNAseq) as it is compatible with many methods of sample preservation, including frozen OCT blocks. To assess if both library modalities were comparable in the context of RNA editing, we first compared RNA editing metrics and cell recovery rates across platforms. Although there are differences in total number of captured RNA editing sites (**Response Figure 6A,B**) and cell recovery rates (**Response Figure 6C**), RNA editing rates captured in matching across single cell and single nucleus libraries correlate strongly ($R=0.811$) with one another (**Response Figure 6D**).

We would like to highlight however that in the manuscript, we never make direct comparisons across datasets generated using different single cell library generation modalities. In the manuscript’s **Figure 4: “Single Nuclei RNA Editing Analysis Across Teratoma**

Development”, we never compare single cell versus single nucleus RNA sequencing modalities. Every comparison in our study is made between two single nucleus RNA sequencing datasets.

Response Figure 6: (A-B) The number of captured RNA editing sites between single cell and single nucleus library preparation modalities, across teratoma cell-types. Sites are reported as either being from repetitive DNA sequences **(A)** or their annotated genetic region **(B)**. **(C)** The number of captured cells between single cell and single nucleus library preparation modalities, across teratoma cell-types. **(D)** The correlation in RNA editing rates between matching sites captured in single cell and single nucleus RNA sequencing libraries.

Minor comments:

7. The nomenclature for ADAR1, ADARB1, and ADARB2 is inconsistent, mixing non-official (i.e., ADAR1) and official (i.e., ADARB1 and ADARB2) protein names. It is recommended to use either "ADAR1, ADAR2, and ADAR3" or "ADAR, ADARB1, and ADARB2" consistently.
8. Several panels are not referenced in the main text (i.e., Fig. 2H, Fig. 3D, Fig. S1C, and Fig. 4B) and should be appropriately cited.
9. In the introduction section, the statement that "A-to-I RNA editing leads to the incorporation of inosines" should be revised to accurately reflect that A-to-I editing results in the deamination of adenosines, converting them to inosines.
10. In bulk human organs, the number of datasets (n) for each organ should be indicated.

Thank you – these changes have been incorporated in the text to enhance clarity.

Reviewer #2 (Remarks to the Author):

The manuscript by Dailamy et al. titled "Charting and Probing the Activity of ADARs in Human Development and Cell-Fate Specification" presents a comprehensive study on the role of Adenosine Deaminases acting on RNA (ADARs) in human development, focusing on their functions in early cell fate specification. The research is well-structured and contributes to the field of developmental biology and gene regulation. The main strengths:

- 1) The multi-pronged methodology combining time-course RNA editing profiles in human organs and hPSC differentiation models is commendable. This approach provides a holistic view of ADARs' roles across developmental stages.
- 2) The utilization of both brain organoids and teratomas for experimental probing of ADARs is technically sound. This dual-model approach enhances the validity of the findings.
- 3) The thorough analysis of RNA editing levels, ADAR enzyme expression dynamics, and single-cell RNA editing profiles is well executed. The use of advanced analytical methods, such as single-cell CRISPR-KO screens, adds robustness to the findings.
- 4) The discovery of ADAR1's role in mesenchymal differentiation and its potential implication in obesity-related phenotypes is a significant contribution to the field.

Conclusion: This manuscript sheds new light on understanding the role of ADARs in human development. The comprehensive approach, rigorous data analysis, and novel insights pave the

way for future research in this area. However, some areas require further improvement before the manuscript can be accepted for publication:

1) While the study focuses on the developmental aspect and provides a discussion on the implications of ADARs in certain tissue/organ development and diseases, not a single edited target gene was mentioned (discovered) to offer deeper insights into the molecular players downstream ADARs whose editing defects may contracture to the observed/speculated functional readouts. Additional work is needed to put ADARs and their edited target genes into the context of the underlying biology.

Thank you for the note. We have now performed a deeper analysis on the time series fetal-to-adult organ RNAseq dataset, in which the prenatal-to-postnatal shift in site-specific RNA editing was measured across all measured sites. These differential editing levels were also correlated with the prenatal-to-postnatal shift in gene expression of the corresponding genes.

More specifically, we employed a site-specific RNA editing pipeline. This pipeline identified edited mRNA sites within the REDportal table across all bulk organ time-series data (forebrain, hindbrain, liver, heart, kidney, and testis). Subsequently, we examined differential editing levels in these sites between postnatal and prenatal samples using customized R scripts, akin to established computational methodologies (doi.org/10.1016%2Fj.celrep.2022.111585). The resulting delta editing values (postnatal vs prenatal) were then correlated with the log fold change in expression levels of corresponding genes between postnatal and prenatal samples (**Response Figure 1**).

Response Figure 1: Schematic depicting the site-specific RNA editing analysis pipeline for developmental bulk organ samples. After bulk organ samples were grouped into prenatal and postnatal groups, differential editing analysis was conducted between these two developmental groups. Finally, prenatal versus postnatal differences in RNA editing levels (delta editing rate) were correlated with corresponding gene expression level changes (log fold-change).

After identifying RNA editing sites at a site-specific level, we next classified editing sites into their corresponding genetic regions, and found that the majority of identified sites are located in 3'UTRs (**Response Figure 2A**). Furthermore, we found that the largest number of editing sites were identified in the hindbrain (**Response Figure 2A**).

Next, we identified sites that are differentially edited between prenatal and postnatal samples across all tissues (**Response Figure 2B**). The delta editing rate of these differentially edited sites were then correlated with the log fold change (logFC) of the corresponding gene expression level changes (**Response Figure 2C**). Across all organs, we observed a slightly negative correlation ($R = -0.133$) between the change in editing levels and gene expression across all identified differentially edited genes (**Response Figure 2C**).

Response Figure 2: (A) Total number of RNA editing sites discovered across all examined organs. (B) Total number of prenatal versus postnatal differentially edited RNA editing sites discovered across all examined organs (n.d = none detected). (C) Prenatal versus postnatal differences in RNA editing levels (delta editing rate/gene) were correlated with corresponding gene expression level changes (log fold-change/gene) for all genes captured across all organs. Delta editing rates per gene were averaged across all delta values for each site per unique gene. Correlation was measured using a Pearson's Correlation Coefficient and 95% confidence intervals are plotted for each genetic region.

Intrigued by this slightly negative correlation, we then separated the data at the organ level, and mostly observed weak trends across the forebrain, hindbrain, kidney and testis tissues (**Response Figure 3A-D**). Interestingly, we noticed a slightly more convincing negative correlation in liver tissues (**Response Figure 3E**), and when separating out the sites at a genetic region level, we noticed a strong negative ($R = -0.544$) correlation in liver intronic editing sites (**Response Figure 3F**). This negative correlation prompted a functional characterization of these genes. Using the ToppGene Suite, we observed GO terms associated with cholesterol biosynthesis and metabolism (**Response Figure 3G**). The input genes that are associated with these cholesterol GO terms are SREBF1, HMGCS1, FDPS, FADS2, and EBP, and to our knowledge there are no literature that explores the effects and dynamics of RNA editing in these cholesterol genes during the fetal-to-adult transition. A further exploration into the significance of RNA editing at these sites may potentially shed light on novel cholesterol gene regulation modalities.

Response Figure 3: (A-F) Prenatal versus postnatal differences in RNA editing levels (delta editing rate/gene) were correlated with corresponding gene expression level changes (log fold-change/gene) for all sites captured in the forebrain **(A)**, hindbrain **(B)**, kidney **(C)**, testis **(D)**, liver **(E)**, and liver intronic sites only **(F)**. **(G)** GO term functional annotation of differentially edited liver intronic sites.

Next, we delved into the RNA editing sites identified through the site-specific RNA analysis pipeline on the time-series bulk organ data. After compiling a list of differentially edited sites between prenatal and postnatal stages across all tissues, we investigated the convergence of these sites across organs (**Response Figure 4A**). This exploration aimed to uncover genes with differentially edited sites common to multiple tissue types, shedding light on potential developmentally conserved RNA editing mechanisms.

We observed 58 prenatal versus postnatal differentially edited sites that were shared across all organ datasets (**Response Figure 4B**). Intrigued by this convergence, we conducted functional annotation on these genes using the ToppGene Suite and identified GO terms associated with innate immunity and DNA replication (**Response Figure 4C**). Notably, EIF2AK2 (also known as PKR) and MAVS, involved in viral response and innate immunity pathways, were found to have differentially edited sites in all analyzed tissues (**Response Figure 4D**). Thus, through an unbiased look at the convergence of prenatal versus postnatal differentially edited RNA editing sites across organs from all three germ-layers, we identified that there is a significant shift in RNA editing of innate immunity related genes during the fetal-to-adult transition.

Response Figure 4: (A) Schematic depicting the site-specific RNA editing analysis pipeline for developmental bulk organ samples. After bulk organ samples were grouped into prenatal and postnatal groups, differential editing analysis was conducted between these two developmental groups. Finally, a convergence analysis was conducted to find shared and unique differentially edited sites between organs. **(B)** Venn diagram illustrating the number of shared and unique prenatal versus postnatal differentially edited sites across all examined organ datasets. **(C)** GO term functional annotation of the pan-organ shared differentially edited sites **(D)** List of the input genes that are associated with these corresponding convergent GO terms.

To assess how shifts in RNA editing correlate with gene expression, we utilized Gene Set Variation Analysis (GSVA), which condenses information from gene expression profiles to calculate gene scores for specific biological pathways. Specifically, we computed gene set scores for C2 gene sets from the Human MSigDB Collections, tracking them over time for the Viral Infection Gene Set (red line) and DNA Replication Gene Set (green line) (**Response Figure 5A-F**). The global editing level of these samples, measured by the Alu Editing Index (AEI), was overlaid onto the graphs to visualize changes in global editing levels across development for each organ (**Response Figure 5A-F**). We observed significant inverse correlations between the AEI and gene scores from these two pathways in the forebrain, hindbrain, and liver (**Response Figure 5A-F**). This analysis underscores the crucial role of A-to-I editing in regulating gene expression in innate immunity and DNA replication pathways during the fetal-to-adult transition.

Response Figure 5: (A-F) Charting global A-to-I editing dynamics (AEI), along with the Viral Infection and DNA replication pathway gene scores, across developmental time, for the forebrain (A), hindbrain (B), heart (C), liver (D), kidney (E), and testis (F). Correlation between the gene score sets and the AEI were measured using the Pearson correlation coefficient.

2) The interpretation of the data, especially in complex systems like human development, is crucial. Ensuring that the conclusions drawn are supported by the data and validated by independent methods or data sets is essential for robustness. The findings from human teratoma and organoids should be validated in the human PSC differentiation model along specific

lineages, even better and most relevant, using human synthetic embryos such as hPSC-derived blastoids.

Our main finding from ADAR-KO teratoma enrichment analysis is that we observe adipogenic enrichment following *ADAR*-KO. To validate the observed increase in Adipogenic-MSCs within teratoma cells following *ADAR*-KO, we conducted experiments using primary human MSCs in vitro. These cells were transduced with lentivirus carrying *ADAR*-targeting shRNA (LV-sh*ADAR*) or a non-targeting shRNA control (LV-shControl) (**Manuscript Figure 8H**). We first verified the effectiveness of *ADAR* knockdown after sh*ADAR* transduction at the onset of adipogenic differentiation (day 0) (**Manuscript Figure 8J**). Furthermore, we observed enhanced adipogenesis at day 8 of adipogenic differentiation in sh*ADAR* samples, as indicated by an upregulation of adipogenic markers *ADIPOQ* and *PPARG* (**Manuscript Figure 8K**) and a significant increase in Oil Red O positive percent cells compared to the shControl (**Manuscript Figures 8I and 8L**). These findings validate the observed adipogenic enrichment in *ADAR*-KO teratomas.

During this review period, we have now further validated this finding using an iPSC-derived MSC differentiation model. Specifically, two distinct iPSC lines (KOLF2.1Js and PGP1s) were differentiated to MSCs and then similarly transduced with LV-sh*ADAR*, along with LV-shControl (**Response Figure 7A**). As seen in adult MSCs, following *ADAR* repression (**Response Figure 7B**) we observed increased adipogenic differentiation, as measured by Oil O Red positive staining following (**Response Figure 7C**).

Response Figure 7: (A) Schematic depicting experimental workflow for transducing iPSC derived MSCs with lentivirus encoding shRNA targeting *ADAR* gene expression, and subsequently checking for its effect on adipogenesis **(B)** *ADAR* and adipogenic transcription factor *CEBPB* transcript quantification in iPSC derived MSCs 5 days post shRNA transduction (n=2). **(C)** Quantification of oil o red staining spots in iPSC derived MSCs cultures 5 days post transduction (n=3).

RESPONSE TO REVIEWER COMMENTS

Reviewer #1 (Remarks to the Author):

The authors have addressed all the points raised by both reviewers. The manuscript now includes additional data with novel conclusions on tissue-specific and pathway-enriched edited targets which improves the manuscript, in the opinion of the reviewer.

Thank you!

Reviewer #2 (Remarks to the Author):

The authors have done a commendable job of revising and improving the manuscript. The study now presents some interesting findings and resourceful datasets that will benefit the scientific community. The manuscript can be accepted for publication with the remaining minor points addressed.

1. Page 4: Fig. 2F, 2H: the AEI upregulation was mentioned to coincide with ADARB1 downregulation in the Kidney. But, how AEI got upregulated was not mentioned or discussed. Did the authors check other ADAR members?

This is an interesting point brought up by the reviewer. The reviewer notes that during the late gestation to newborn-teenager phase of kidney development, we see an increase in AEI. Intriguingly, all ADAR member expression was assessed and besides a downregulation of ADARB1 expression during this developmental shift, there were no other significant trends in expression level changes.

2. Page 6: "Innate Immunity" GO term was mentioned in the result text for Fig. 3C; however, Fig. 3C did not present the "innate Immunity" GO term.

Our GO term analysis did not present a GO term that is precisely called "innate immunity" and it instead presented 3 GO terms in the top 5 terms that are related to innate immunity activation. This is why, in the main text, we say: "... we conducted functional annotation on these genes using the ToppGene Suite and identified GO terms *associated* with innate immunity and DNA replication (Figure 3C)." We further explain that "EIF2AK2 (also known as PKR) and MAVS, involved in viral response and innate immunity pathways, were found to have differentially edited sites in all analyzed tissues." These two are proteins are known to play essential roles in the innate immune response to viral infection (<https://pubmed.ncbi.nlm.nih.gov/10933401/> ; <https://www.frontiersin.org/journals/immunology/articles/10.3389/fimmu.2020.01030/full>).

3. Page 6: AEI (blue line) should be pointed out in the results text before (Fig. 3E-J) to be consistent with the references to the other two colored lines mentioned already.

This is a great point and was added to the main text to enhance clarity.

4. Page 7: “Supplementary Fig. 3” was referred to with a single summary statement without any meaningful explanation or mention of all the 10 (A-J) figure panels. What those 10 figure panels are should be explained.

This is a great point that will help enhance reader clarity; thus, the following block was added to the main text:

“Lastly, we conducted a similar teratoma generation pipeline on 3 other commonly used PSC cell lines (H9s, HUES62s, and PGP1s, **Supplementary Figure 3A**). We were able to call all major cell-types in H9 teratomas (**Supplementary Figure 3B**) and saw similar AEI levels (**Supplementary Figure 3C**) and AEI-to-ADAR correlation levels (**Supplementary Figure 3D**) as well. These observations and trends were also seen in the HUES62 (**Supplementary Figure 3E-G**) and PGP1 (**Supplementary Figure 3H-J**) cell lines.”

5. Page 9: “cosine similarity index” was mentioned and referred to (Fig. 5F), which was not present in that Fig. 5F panel.

Cosine similarity index is indeed not used in the analysis done in Figure 5F. This was clarified in the text.